# DATA VALUATION FOR GRAPHS

## ABSTRACT

What is the worth of a node? We answer this question using an emerging set of data valuation techniques, where the value of a data point is measured via its marginal contribution when added to the (training) dataset. Data valuation has been primarily studied in the i.i.d. setting, giving rise to methods like influence functions, leave-one-out estimation, data Shapley, and data Banzhaf. We conduct a comprehensive study of data valuation approaches applied to graph-structured models such as graph neural networks in a semi-supervised transductive setting. Since all nodes (labeled and unlabeled) influence both training and inference we construct various scenarios to understand the diverse mechanisms by which nodes can impact learning. We show that the resulting node values can be used to identify (positively and negatively) influential nodes, quantify model brittleness, detect poisoned data, and accurately predict counterfactuals[1].

## 1 INTRODUCTION

When we learn from data, a natural question to ask is *how each data point influences learning*. We can study how a given training point affects the learned weights, the accuracy, or the prediction for a given test point. Beyond offering insights into our models, this question is also practically important, especially for modern machine learning methods that rely on increasingly larger datasets (Zhou et al., 2017). For example, if we gather data from unreliable sources (e.g. the internet) we may want to filter out low quality instances. When we obtain data from data markets, different data providers (e.g. individuals, hospitals) should be equitably compensated for the data that they provide. Data values have been used to quantify train-test leakage, find semantically similar examples (Ilyas et al., 2022), detect mislabeled instances (Wang & Jia, 2023) and for dataset selection (Engstrom et al., 2024).

A classical answer to the data valuation question is influence functions (Cook & Weisberg, 1982). They approximate the effect of removing a data point on the model performance. From linear regression to deep learning, they have been thoroughly studied in various settings, including their limitations (Basu et al., 2021; Bae et al., 2022). Broadly, data valuation encompasses techniques that try to relate some output of a model with the data it is trained on. While there is growing research that explores different notions of value, the vast majority is focused on supervised learning for i.i.d. data.

We conduct the first comprehensive data valuation study for graph-based models in the semi-supervised transductive setting. Only two recent works have explored data valuation for graphs. Chen et al. (2023) derive closed-form estimate for the leave-one-out influence of nodes and edges using SGC (Wu et al., 2019) as a surrogate model. Chi et al. (2024) propose the precedence-constrained winter value where nodes are grouped in coalitions based on the graph structure. Our focus is not on proposing new notions, but rather rigorously evaluating existing notions. Our results show that both data Banzhaf (Wang & Jia, 2023) and datamodels (Ilyas et al., 2022) significantly outperform other valuation methods, but neither of these notions has been studied for graphs so far. Given the importance and ubiquity of graph-based models such as graph neural networks (Ju et al., 2024), we hope that our work provides a solid foundation for the study of data valuation on graphs.

It is clear that not all data are created equally, e.g. some instances may be noisy or mislabeled. However, even a "high-quality" instance may not be valuable if there are already similar instances in the dataset, indicating that to properly evaluate the worth of an instance we need to carefully consider the appropriate context. Lack of context is why leave-one-out estimation often fails in practice – its focus

---

[1]The code to reproduce results is provided as supplementary material.

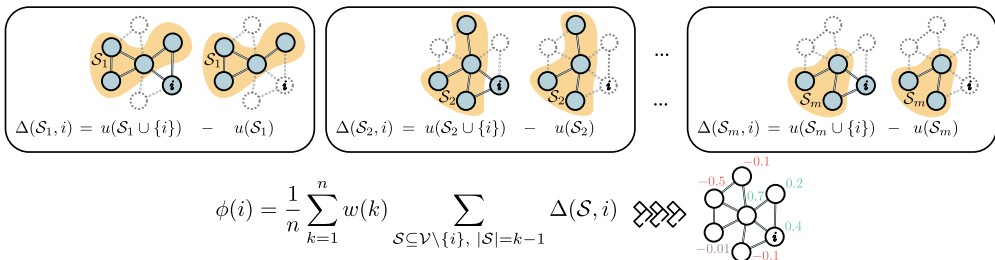

$$\phi(i) = \frac{1}{n} \sum_{k=1}^{n} w(k) \sum_{\mathcal{S} \subseteq \mathcal{V} \setminus \{i\}, \; |\mathcal{S}| = k-1} \Delta(\mathcal{S}, i)$$

Figure 1: Data valuation assigns importance to a node based on its marginal contribution to the utility across various subsets. When removing the node: if model performance decreases, it is considered informative (green); if performance remains unchanged, the node is irrelevant (grey); if performance improves, the node is misleading (red). Weighting schemes $w$ define different data valuation methods.

is on the effect of an instance in isolation. In contrast, recent game-theoretic notions such as data Shapley (Ghorbani & Zou, 2019) and other semi-values (Dubey et al., 1981) compute the average marginal contribution of an instance considering all potential subsets of the data. This indeed comes at a significant computational cost, but approximation strategies allow us to compute useful estimates in practice.

The right context is even more important in the graph setting. First, the prediction of a node depends on its neighborhood – nodes are not i.i.d. This suggests that looking at nodes in isolation is likely suboptimal and we should consider different subgraphs. The main idea is illustrated in Fig. 1. To estimate the influence of node $i$ on some utility function $u(\cdot)$ such as the accuracy, we consider the difference in utility $\Delta(\mathcal{S}, i) = u(\mathcal{S} \cup \{i\}) - u(\mathcal{S})$ with and without node $i$ across many subsets $\mathcal{S} \subseteq \mathcal{V}$ of different sizes. These marginal contributions $\Delta(\mathcal{S}, i)$ are merged together using a weighted average to obtain the final node value $\phi(i)$. Different weights recover many of the established data value notions, including data Shapley, beta Shapley, data Banzhaf, and leave-one-out.

Second, all nodes – labeled and unlabeled – influence both training and inference. We account for this by constructing different variants (§ 3) of each data value notion that aim at disentangling these effects. Unsurprisingly, even though the number of training nodes is a small percentage of all nodes in the sparsely-labeled scenario (Shchur et al., 2018), the top most influential nodes are mostly training nodes. At the same time, we can find unlabeled nodes which have greater influence than many training nodes. This reflects the complex interplay between training and test nodes in graphs.

In addition to influence-based and game-theoretic notions, predictive approaches such as datamodels (Ilyas et al., 2022) and MLPbV (Wu et al., 2024) construct surrogate (proxy) models that can predict the utility function for any subset without the need to train the model from scratch. They allow us to answer counterfactual questions, i.e. predicting the performance on arbitrary unseen subsets of data. Proxy models are trained on subset of all possible $\{(\mathcal{S}_i, u(\mathcal{S}_i)\}$ pairs, where $\mathcal{S}_i \subseteq \mathcal{V}$. As we discuss in § 2.1, we can trivially turn any set of data values into a predictive surrogate. Moreover, the resulting data values from predictive and game-theoretic approaches are equivalent for certain configurations (see § 2.2). Thus, this distinction between these types of notions may not be very useful.

Overall, our results show that approaches accounting for subgraphs instead of single-node contribution result in more accurate data values. We demonstrate the usefulness of the data values for various downstream applications: i) *finding highly influential nodes* – pruning nodes with high positive (negative) influence results in a significant drop (rise) in model performance; ii) *spotting brittle predictions* which depend on a small set of support nodes whose removal results in misclassification; iii) *detecting poisoned (mislabeled) data*; iv) *estimating counterfactuals* such as predicting the performance on an arbitrary subset; and v) *visualizations* that provide further insight in the data.

We thoroughly study the problem of data valuation for graphs in a transductive semi-supervised setting, including variants that attribute importance to both labeled and unlabeled nodes. We analyse different valuation approaches, including game-theoretic and predictive notions. We show that datamodels and data Banzhaf, neither of which was previously considered in the context of graphs, significantly outperform other valuation techniques, including the two recently proposed graph-specific approaches. Our study was computationally expensive (over 2500 compute hours in total) and storage-intensive (more than 10TB of raw data). Nonetheless, we show that one can obtain accurate estimates even with modest computational resources, and we plan to publicly release the raw data.

## 2 DATA VALUATION

Broadly, data valuation methods try to relate the presence or absence of an instance in the dataset with the resulting change in some user-defined utility function. Traditionally, this is some measure of model performance such as accuracy, where a high positive (negative) value indicates that including the instance in the data improves (deteriorates) performance. For both game-theoretic and predictive approaches the main idea is to consider different subsets of the training set $\mathcal{D}$ and assess the value of missing training samples according to the change in the utility. We denote with $\phi(i) \in \mathbb{R}$ the value assigned to instance $i$.

**Utility.** Let $u : 2^{|\mathcal{D}|} \to \mathbb{R}$ be a utility function which maps any subsets $\mathcal{S} \subseteq \mathcal{D}$ to a score indicating the usefulness of the subset. For classification tasks, we commonly have $u(\mathcal{S}) = \text{acc}(\mathcal{A}(\mathcal{S}))$ where $\mathcal{A}$ is a learning algorithm that returns a model $f_{\mathcal{S}}$ trained on $\mathcal{S}$, and acc computes the accuracy on a held-out set. Sometimes we are not interested in the overall performance but rather how the training data influences a specific test point. For example, the utility can be the prediction margin for the test point $\boldsymbol{x}_{\text{test}}$. That is $u(\mathcal{S}) = u(\mathcal{S}, \boldsymbol{x}_{\text{test}}) = \text{margin}(f_{\mathcal{S}}, \boldsymbol{x}_{\text{test}}) = f_{\mathcal{S}}(\boldsymbol{x}_{\text{test}})_{y^*} - \max_{y \neq y^*} f_{\mathcal{S}}(\boldsymbol{x}_{\text{test}})_y$ where $f_{\mathcal{S}}(\cdot)_y$ returns the logit-score or probability for label $y$, and $y^*$ is the ground-truth label. Any other function of (the outputs of) $f_{\mathcal{S}}$ is valid.

### 2.1 CATEGORIZATION OF DATA VALUATION APPROACHES

**Game-theoretic notions.** These methods consider different subsets (coalitions) of the training data, treating each data point as a player in a cooperative game. If $u(\mathcal{D})$ represents the outcome of the game, the goal is to fairly distribute the outcome to each data point according to its contribution. This concept is at the core of Shapley values (Shapley, 1953; Ghorbani & Zou, 2019), a well-known framework for assessing the value of data according to the average marginal contribution of adding a data point into all possible subsets, weighted by the number of permutations in which that data point appears.[2] Mathematically, the Shapley value of a data point $i$ is given by

$$\phi_{\text{SHAP}}(i) = \frac{1}{n} \sum_{k=1}^{n} \binom{n-1}{k-1}^{-1} \sum_{\mathcal{S} \subseteq \mathcal{D} \setminus \{i\}, |\mathcal{S}|=k-1} [u(\mathcal{S} \cup \{i\}) - u(\mathcal{S})] \tag{1}$$

To compute it exactly we need to train $2^{|\mathcal{D}|}$ models – one for each possible subset. Since this is prohibitively expensive there are various approximation techniques such as Monte Carlo sampling. Shapley values are popular since they uniquely satisfy four axioms that were argued to be necessary to ensure a fair valuation. These are *symmetry*, *linearity*, *null player*, and *efficiency*.

Kwon & Zou (2022) questions the necessity of the *efficiency* axiom, which requires all the values to sum up to the utility on the original dataset, i.e. $\sum_{i \in \mathcal{D}} \phi(i) = u(\mathcal{D})$. Removing this axiom we get the so called *semivalues* (Dubey et al., 1981) which satisfy the other three axioms. As with any axiomatic approach, these axioms can be debated. For the purposes of this paper, we consider them as given. It turns out that all semivalues can be written in a canonical form. Let $n = |\mathcal{D}|$ then

$$\phi_{\text{semivalue}}(i, w) = \frac{1}{n} \sum_{k=1}^{n} w(k) \sum_{\mathcal{S} \subseteq \mathcal{D} \setminus \{i\}, |\mathcal{S}|=k-1} [u(\mathcal{S} \cup \{i\}) - u(\mathcal{S})] \tag{2}$$

where $w : [n] \to \mathbb{R}$ is a weight function such that $\sum_{k=1}^{n} \binom{n-1}{k-1} w(k) = n$.

We can recover Shapley (SHAP) with $w(k) = \binom{n-1}{k-1}^{-1}$. Leave-one-out (LOO) measures the difference in utility when removing a single data point from the dataset and can be obtained by choosing $w(k) = n\mathbf{1}[k = n]$. Data Banzhaf (BANZ) assigns uniform weights $w(k) = \frac{n}{2^{n-1}}$.

**Predictive notions.** Ilyas et al. (2022) introduce the datamodel framework where the idea is to learn a surrogate model $m_{\boldsymbol{\theta}} : \{0, 1\}^{|\mathcal{D}|} \to \mathbb{R}$ that can directly predict the utility for any given subset, i.e. $m_{\boldsymbol{\theta}}(\mathcal{S}) \approx u(\mathcal{S})$. The weights $\boldsymbol{\theta}$ of the surrogate are learned from $\{(\mathcal{S}_i, u(\mathcal{S}_i))\}$ where $\mathcal{S}_i$ is encoded

---

[2]The initial use of Shapley values in the machine learning community was for data attribution – determining which features (e.g. pixels) highly influence the prediction. That is, they were used as a post-hoc explainability method. In contrast, Ghorbani & Zou (2019) introduce data Shapley where the goal is to consider the influence of training points instead, which is the setting that we consider in this paper (see also § G).

as a binary vector $\{0, 1\}^{|\mathcal{D}|}$ indicating the presence or absence of an instance. They show that even a simple linear model can approximate the mapping from $\mathcal{S}$ to $u(\mathcal{S})$ well. The data values are given by the weights of the linear model $\boldsymbol{\theta} \in \mathbb{R}^{|\mathcal{D}|}$, i.e. $\phi_{\text{DM}}(i; \boldsymbol{\theta}) = \theta_i$. Wu et al. (2024) propose to instead use a multi-layer perception as a surrogate which can use additional inputs.

**Influence functions.**     The number of possible subsets increases exponentially with the size of the dataset. Moreover, for each subset, we need to train the entire model from scratch. If the dataset is large even computing the leave-one-out error which requires "only" $n = |\mathcal{D}|$ evaluations can be too expensive. In contrast, influence functions (Cook & Weisberg, 1982) aim to approximately compute the change in the model output (or model weights) when removing a training sample. They rely on first and second-order information (gradient, Hessian) to obtain estimates without model retraining. Nevertheless, influence functions perform poorly in non-convex settings as in deep neural networks (Basu et al., 2021; Bae et al., 2022). At best, they perform as well as leave-one-out errors which are suboptimal since they consider instances in isolation. Chen et al. (2023) derive different data values using influence functions for graph neural networks.

**From game-theoretic to predictive.**     Any game-theoretic approach can be made predictive by assuming that the values are the weights of an implicit linear model (without a bias term). This implies that the predicted value of any unseen subset equals the sum of the data values of the nodes in the subset, i.e. $u(\mathcal{S}) \approx \sum_{i \in \mathcal{S}} \phi(i)$ for any $\phi(i)$. This is a reasonable assumption given the linearity axiom of semivalues, and it's explicitly made by datamodel values (Ilyas et al., 2022).

**Sampling subsets.**     Since it is infeasible to evaluate the utility for all $2^{|\mathcal{D}|}$ subsets a common approximation technique is to sample subsets. A simple but effective choice is to independently include each instance with some probability $\alpha$, that is $\Pr[i \in \mathcal{S}] = \alpha, \forall i \in \mathcal{D}$. This implies that the subset size follows a Binomial distribution with an expected set size of $\alpha \cdot |\mathcal{D}|$. We follow this approach for data Banzhaf and datamodel values. As we show in § E.4, $\alpha$ is an important hyperparameter to tune. To approximate the Shapley value we first randomly sample a permutation, and then scan the nodes in order until reaching a user-defined truncation threshold (see § 4). For a fair comparison, we make sure that each method uses the same number of subsets.

## 2.2   Properties of data valuation approaches

**Connections between different values.**     If $\alpha = 0.5$ and we use no regularization when fitting $m_{\boldsymbol{\theta}}$ the datamodel values are equivalent to Banzhaf values since the Banzhaf values are the best linear approximation to $u$ in terms of least square loss (Wang & Jia, 2023; Hammer & Holzman, 1992). Lin et al. (2022) prove that if we sample $\alpha \sim \text{Unif}(0, 1)$ rather than having it fixed, then the optimal weights of a certain regularized linear model converge relatively quickly to the Shapley values as we increase the number of samples.

**Limitations of data values.**     The utility function can be viewed as a vector in $\mathbb{R}^{2^n}$ which is transformed into a vector of data values $\phi \in \mathbb{R}^n$. This mapping is not injective, so there are different utility functions that yield identical values. A natural question is which utility functions are well approximated by data values. Wang et al. (2024) show that a sufficient condition is for the utility to be a monotonically transformed modular function. In general, data values may be no better than a random baseline for dataset selection. They also generalize the result from Saunshi et al. (2022) which states that the the residual error of the unregularized linear approximation equals to the sum of all $u$'s Fourier coefficients of order 2 or higher. Nonetheless, data values work well in practice. This is remarkable since they approximate $u(\mathcal{S})$ which involves training of a model on $\mathcal{S}$ from scratch (often a complicated neural network) and then computing some function of the model outputs.

**Robustness and efficiency.**     In our context the utility function is stochastic due to the randomness in the training process, making the influence of a data point a random variable with an expectation and a variance (Nguyen et al., 2024). Thus, using a single sample of $u(\mathcal{S})$ may be unreliable. Wang & Jia (2023) define the safety margin – the largest noise that a semivalue can tolerate without altering the ranking of any distinguishable pair of data points. They prove that the data Banzhaf achieves the largest safety margin among all safety values. This property can be attributed to the uniform weights since $\phi_{\text{BANZ}}(i) = \mathbb{E}_{\mathcal{S} \sim \text{Unif}(2^{n \setminus i})}[u(\mathcal{S} \cup \{i\}) - u(\mathcal{S})]$. The standard Monte Carlo estimator directly samples $\mathcal{S}$ from $\text{Unif}(2^{n \setminus i})$ but needs to be repeated $n$ times once for each $i$. Wang & Jia (2023) propose an alternative estimator where given samples $\mathcal{M} = \{\mathcal{S}_1, \ldots, \mathcal{S}_m\}$ i.i.d. from $\text{Unif}(2^n)$ it

computes $\phi_{\text{BANZ}}(i) = \frac{1}{|\mathcal{S}_{\in i}|} \sum_{\mathcal{S} \in \mathcal{S}_{\in i}} u(\mathcal{S}) - \frac{1}{|\mathcal{S}_{\notin i}|} \sum_{\mathcal{S} \in \mathcal{S}_{\notin i}} u(\mathcal{S})$ as the difference of the average utility of the sets $\mathcal{S}_{\in i} \subseteq \mathcal{M}$ that contain $i$, minus the sets $\mathcal{S}_{\notin i} \subseteq \mathcal{M}$ that do not contain it. Now, maximal sample reuse (MSR) is achieved since all evaluations of $u(\cdot)$ are used in the estimation of all $\phi_{\text{BANZ}}(i)$ values. The existence of an efficient MSR estimator is a unique to the Banzhaf value among all semivalues. Nonetheless, we apply the same idea when instances in the subsets are sampled with a fixed probability $\alpha$ (rather than uniform). This actually establishes a new semivalue that we refer to as $\alpha$-`BANZ`, whose weighting scheme is discussed in § B. In practice $\alpha$-`BANZ` outperforms `BANZ` and shows strong results across all settings.

## 3    DATA VALUATION ON GRAPHS

**Semi-supervised setting.**    Consider a graph $\mathcal{G} = (\mathcal{V}, \mathcal{E})$ where $\mathcal{V}$ is the set of nodes and $\mathcal{E}$ is the set of edges. Let $\boldsymbol{X} \in \mathbb{R}^{|\mathcal{V}| \times d}$ be the matrix of $d$-dimensional node features, $\boldsymbol{A} \in \{0, 1\}^{|\mathcal{V}| \times |\mathcal{V}|}$ be the adjacency matrix and $\boldsymbol{y} \in \mathbb{R}^{|\mathcal{V}|}$ be the vector of labels. Let $\mathcal{V}_\ell$ and $\mathcal{V}_u$ be the subsets of labeled and unlabeled nodes respectively, and let $\mathcal{V}_\ell$ be further split into training nodes $\mathcal{V}_t$ and validation nodes $\mathcal{V}_v$. Usually, $\mathcal{V}_\ell$ is a small subset of nodes while the majority is in $\mathcal{V}_u$ (Shchur et al., 2018). In a transductive semi-supervised learning setting, the model is exposed to both $\mathcal{V}_\ell$ and $\mathcal{V}_u$ during the training, such that it can propagate label information from the few labeled nodes to the many unlabeled nodes leveraging the underlying graph connectivity and node features.

**Graph Neural Networks (GNNs).**    In each layer $k$ of a GNN, the hidden representation $\boldsymbol{h}_v^{(k)}$ of a node $v$ is the result of aggregating information from its neighborhood $\mathcal{N}(v)$ and its own representation from the previous layer. Many popular GNNs can be succinctly written in matrix notation as $\boldsymbol{H}^{(k)} = \sigma\left(\boldsymbol{S}\boldsymbol{H}^{(k-1)}\boldsymbol{W}^{(k)}\right)$, where $\sigma$ is the non-linearity, $\boldsymbol{W}^{(k)}$ are trainable parameters, $\boldsymbol{S}$ is the spatial graph convolution operator and $\boldsymbol{H}^{(0)} = \boldsymbol{X}$. For example, in GCN (Kipf & Welling, 2017) we use the degree normalized adjacency matrix $\boldsymbol{S} = \tilde{\boldsymbol{D}}^{-\frac{1}{2}} \tilde{\boldsymbol{A}} \tilde{\boldsymbol{D}}^{-\frac{1}{2}}$, where $\tilde{\boldsymbol{A}} = \boldsymbol{A} + \boldsymbol{I}_{|\mathcal{V}|}$ and $\tilde{\boldsymbol{D}}_{ii} = \sum_j \tilde{\boldsymbol{A}}_{ij}$ is the degree matrix. SGC (Wu et al., 2019) uses the same architecture as GCN without the non-linearity resulting in a model that is linear w.r.t. the weights.

**Training Nodes vs. All Nodes.**    In the transductive setting, all nodes influence learning. The training nodes do so directly since the loss is computed using their ground-truth labels as supervision signal. The remaining nodes do so indirectly since they influence the hidden representation of the training nodes via the graph structure. This means that in addition to investigating the value of the training nodes $\mathcal{V}_t$ as in the standard i.i.d. setting, we can also compute the value of all nodes $\mathcal{V}$. We refer to these two settings as `train` and `all` respectively. Let $\mathcal{D}$ be the set of considered nodes, namely either $\mathcal{D} = \mathcal{V}_t$ or $\mathcal{D} = \mathcal{V}$. Recall that to compute data values we consider different subsets $\mathcal{S} \subseteq \mathcal{D}$. We first remove the nodes that are in $\mathcal{D}$ but not in $\mathcal{S}$, and then we train a GNN on the graph induced by the remaining nodes $\mathcal{T} = \mathcal{V} \setminus (\mathcal{D} \setminus \mathcal{S})$. In the `train` setting

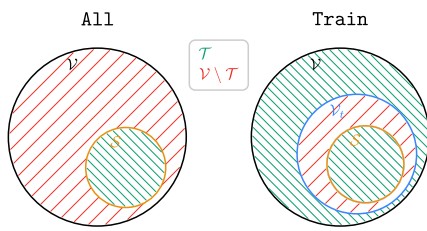

Figure 2: A Venn diagram of the subsets used for the induced subgraph. Hatches slanting downward from left to right indicate the final subset of nodes.

$\mathcal{T} = \mathcal{V} \setminus (\mathcal{V}_t \setminus \mathcal{S})$ and in the `all` setting $\mathcal{T} = \mathcal{V} \setminus (\mathcal{V} \setminus \mathcal{S}) = \mathcal{S}$. Fig. 2 shows Venn diagram representations of the considered subgraphs in the two settings. Since all nodes influence learning we focus on the `all` setting in the main paper. However, we provide additional results with the `train` setting in § F.2.

**Learning signal vs. Overall signal.**    After training, we need to evaluate our utility function. We consider the *accuracy* to measure the influence on the final performance, and the *prediction margin* to measure influence w.r.t. any individual node (see § 2). Here we assume access to ground-truth test labels since our goal is to understand how nodes influence learning for different models. This is standard in the data valuation literature. In practical scenarios, one can use validation labels instead. Importantly, after training on the graph induced by $\mathcal{T}$ we can compute the utility in two different ways which provide different and complementary insights. In particular, we can compute the utility on the same induced subgraph $\mathcal{T}$ which captures the `overall` influence of a node since removed nodes are not present during inference. Alternatively, we can compute the utility on the whole graph $\mathcal{V}$ – that is we bring back the removed nodes $\mathcal{V} \setminus \mathcal{T}$ which only captures their influence during learning.

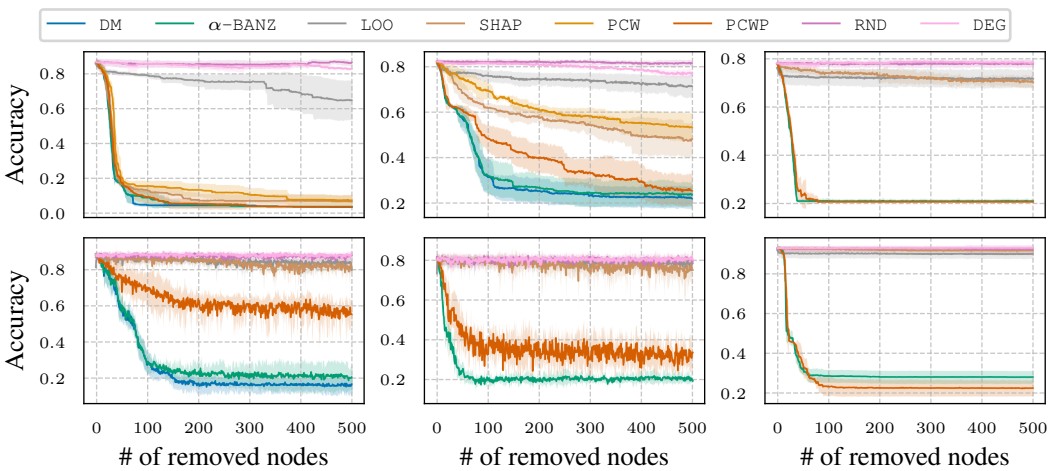

Figure 3: Influential node removal. From left to right, `Citeseer`, `CoraML`, and `PubMed` (first row), and `Photo`, `Computers`, and `CoPhysiscs` (second row).

These `learning` data values measure the training signal provided by a node, while the `overall` data values measure the influence during training and inference. This is in contrast to the i.i.d. setting where we can only measure the train signal provided by an instance. See § E.1 for more details.

## 4 EXPERIMENTAL RESULTS AND ANALYSIS

We compare different data values in a diverse range of applications. Unless specified otherwise, the presented results are in the `all` setting and the `learning` signal variant.

**Methods.** As representative game-theoretic notions, we consider the three well-known semivalues – leave-one-out (`LOO`), data Shapley (`SHAP`), and data Banzhaf ($\alpha$-`BANZ`), where $\alpha = 0.5$ recovers the original Banzhaf value. As a representative predictive notion, we consider datamodels (`DM`). Finally, we consider the precedence-constrained winter value (`PCW`) as the only data value custom-designed for graphs since the method by Chen et al. (2023) approximates `LOO`, which we already cover. We introduce a sampling improvement to `PCW` which we call `PCWP` (see § E.3 for details). We include two more baselines, namely degree (`DEG`) where a node value corresponds to its degree, and randomly assigning node values (`RND`). We test these methods on the largest connected component (LCC) of different citation graphs (`Citeseer`, `CoraML`, `PubMed`, and `CoPhysiscs`) and co-purchase graphs (`Photo` and `Computers`). We consider three models SGC, GCN and GAT. Details about datasets, models, and training are provided in § E.

**Subsets sampling.** We sample $50\,000$ subsets for each method. For $\alpha$-`BANZ` and `DM` we explored different values for $\alpha$ (see § 2.1) and determined the optimal value for `CoraML` (see § E.4). This optimal value, $\alpha = 0.1$, is then used across the other graphs. For `SHAP`, we adopt the implementation from Ghorbani & Zou (2019), which stops processing a permutation after having scanned 25% of the nodes. Thus, we set the number of permutations to $\frac{50\,000}{\lceil N \cdot \text{tr} \rceil}$, where $\text{tr} = 0.25$ is the truncation ratio. When computing node values in the `all` setting, permutations may not start with training nodes resulting in injecting noise (resulting from the random model's prediction). Thus, we start updating node values from the first training node in the permutation and truncate after 25% counting from the latter. Similarly, we select the number of permutations for `PCW` s.t. the total number of subsets equals $50\,000$. In Chi et al. (2024) the default ratios $\text{tr}$ for 1-hop and 2-hop neighbors is $0.5$ and $0.7$ respectively. We optimize these ratios to obtain `PCWP` that performs slightly better (see § E.3).

**Node influence.** As the utility computed to assess node values takes into account both node features and its associated edges – because of the message passing – we consider the value of a node as the contribution of these two components. To evaluate the quality of the node values, we can use the rank of the nodes according to their values. Nodes with high positive value should have positive

influence on the model, while nodes with high negative value should have the opposite effect. A standard analysis is to then remove (or add) nodes according to the rank and observe the change in performance. Fig. 3 shows the results for removing high-ranking nodes across datasets. The utility is the accuracy of a GCN model. We expect a steep decline in the performance as we remove the first most important nodes and then a plateau when removing non-influential nodes. We stop after removing 500 nodes since we are only interested in the initial performance drop. Overall, DM and $\alpha$-BANZ show consistently strong results and have a steepest decline. PCWP performs worse, but better than PCW and SHAP. We omit DM on some of the large datasets due to memory issues. LOO and SHAP converge to the simple RND and DEG baselines on larger graphs. In contrast, PCWP seems to work better on larger graphs (e.g. PubMed and CoPhysiscs) but still lags behind DM and $\alpha$-BANZ. In § F we have many additional experiments with all variants, including node addition. The conclusions are similar.

In the first column of Fig. 4 we show how the methods differ for learning (first row) vs. overall (second row) signal values. To show performance across different settings, we test GAT and use test margins as the utility (see § E for details on how to go from margins to a single influence value). Again, here we can see that DM and $\alpha$-BANZ outperform all other methods in both settings. The gap between the methods is larger in the overall setting. Here the valuation methods have "less information" since if node $i \notin \mathcal{S}$ the node is not available during training or inference. In all cases, removing just a few high-ranking nodes is enough to completely destroy the performance.

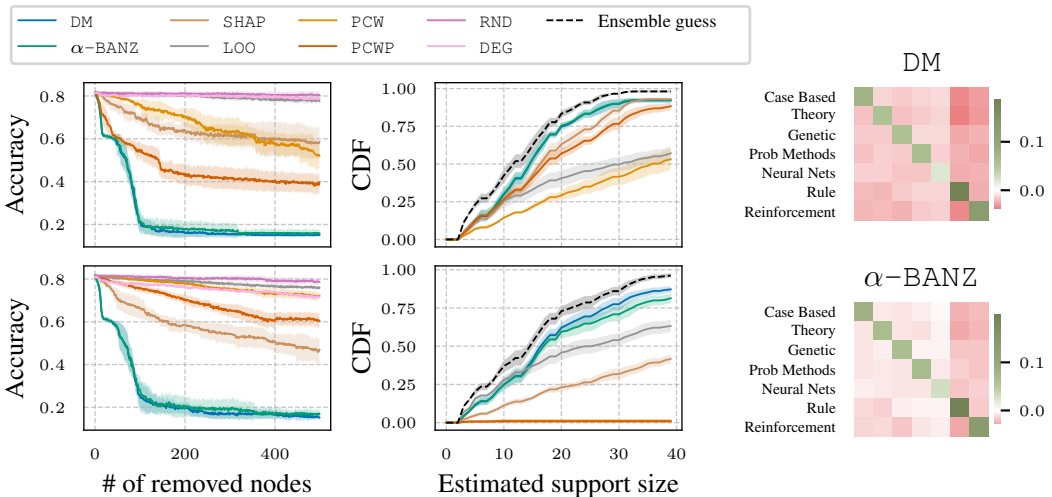

Figure 4: Most influential node removal with a GAT model (left); Figure 5: Average class-wise values CDF of brittle predictions with a SGC model (right). ues with a GCN model.

**Brittle predictions.** The support of a node $v$ is the minimum set of nodes such that if removed, the node $v$ is misclassified. This concept was introduced by Ilyas et al. (2022) for images. Using the margin of $v$ as a utility we can estimate nodes that have high influence on $v$, which in turn allows us to estimate the support. We follow Algorithm 1 from Ilyas et al. (2022), using different data values to estimate the top-$k$ nodes and the support. In the second column of Fig. 4 we see that, regardless of the signal of the values (learning in the first row, and overall in the second) most nodes are brittle – more than half of the nodes (y-axis) have a support size of 15 or less (x-axis). By removing only 15 influential nodes we can misclassify more than $50\%$ of nodes. Computing the exact support is intractable and each data value gives us an upper bound. Choosing the best upper bound among all values we arrive at the best estimate marked with a black dash line (ensemble guess). Again we see that DM and $\alpha$-BANZ are closest to this estimate.

**Average class-wise values.** When the utility is the prediction margin we can compute the influence of every node on every other node. To understand how nodes influence each other we compute the average influence between nodes from each pair of classes and visualize the result in Fig. 5. We see positive data values within classes and negative between classes. Interestingly, the last two classes have a stronger influence on average. More insights on the class-wise influence are provided in § D.3.

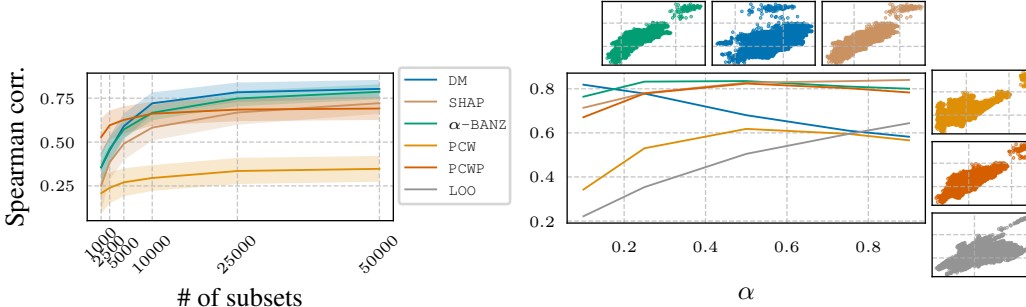

(a) Linear datamodeling score (LDS) with the increasing subsets' number.

(b) Counterfactual predictions across different alphas. Insets show the correlation at $\alpha = 0.5$.

Figure 6: LDS (Spearman correlation) on SGC with prediction margins as the utility.

**Linear datamodeling score (LDS).** To evaluate data values Park et al. (2023) introduce the LDS score – defined as the Spearman correlation between the predicted utility and the true utility. The idea is that a good (predictive) valuation method should accurately approximate the utility on held-out subsets. First, we study how the number of considered subsets influences the LDS score. We compute data values with 90% of the subsets and use the remaining 10% to compute the LDS score. We study SGC with the prediction margin as the utility and report the average results across 5 different seeds. As shown in Fig. 6a, all the approaches benefit from a larger number of considered subsets. However, DM and $\alpha$-BANZ show relatively good performance even with just $5\,000$ subsets.

**Counterfactual performance.** In Fig. 6a the held-out set is sampled from the same distribution that is used to compute the data values (both with $\alpha = 0.1$). Next, we study how the performance (in terms of the LDS score) changes if the distribution of the held-out subsets changes – termed counterfactual sets in Ilyas et al. (2022). To test this we vary the $\alpha$ of the held-out subsets, keeping $\alpha = 0.1$ fixed for computing the data values.

Fig. 6b shows that across different $\alpha$ values, $\alpha$-BANZ generalizes better even though during training it just considered subsets with $\alpha = 0.1$. PCW and PCWP also generalize well, while DM becomes worse for a larger distribution shift (larger $\alpha$). The scatter plot insets show that the predicted utility (x-axis) is correlated with the true utility (y-axis).

**Poisoning.** When gathering data from external sources, some instances may be corrupted intentionally (poisoning) or unintentionally (mislabeling). The rank of node values can help detect such instances. Namely, a high data value of a labeled node for itself (self-importance) indicates memorization. When a training node $v$ has a wrong label and $v \in \mathcal{S}$ the model must memorize this wrong label to achieve low training loss. If $v \notin \mathcal{S}$ then the model's prediction does not match the wrong label. This leads to a large difference $u(\mathcal{S} \cup \{v\}) - u(\mathcal{S})$, and thus a large value. To test this, we poison $10\%$ of the training data and compute the node values using the margin of the nodes themselves as the utility.

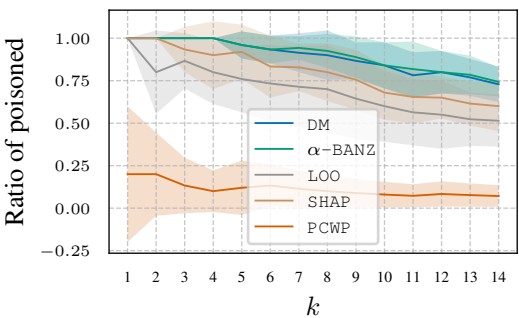

Figure 7: Ratio of poisoned nodes in top-$k$ rank according to node values across 5 different seeds.

We consider the `learning` signal in the `train` setting. In Fig. 7, we show the percentage of poisoned data for `CoraML` appearing in top-$k$ ranked nodes as $k$ increases. DM and $\alpha$-BANZ are consistently better at detecting poisoned data than the others. For more details see § F.3.

**Transferability.** In Fig. 8 we repeat the node removal experiment (similar to Fig. 3) with two variants. We either compute top-$k$ data values with the GCN model and then evaluate the performance on GCN (no transfer, solid line), or we compute top-$k$ data values with SGC and see whether they transfer to GCN (dashed line). We also look at the transfer from SGC to GAT. We consider margins

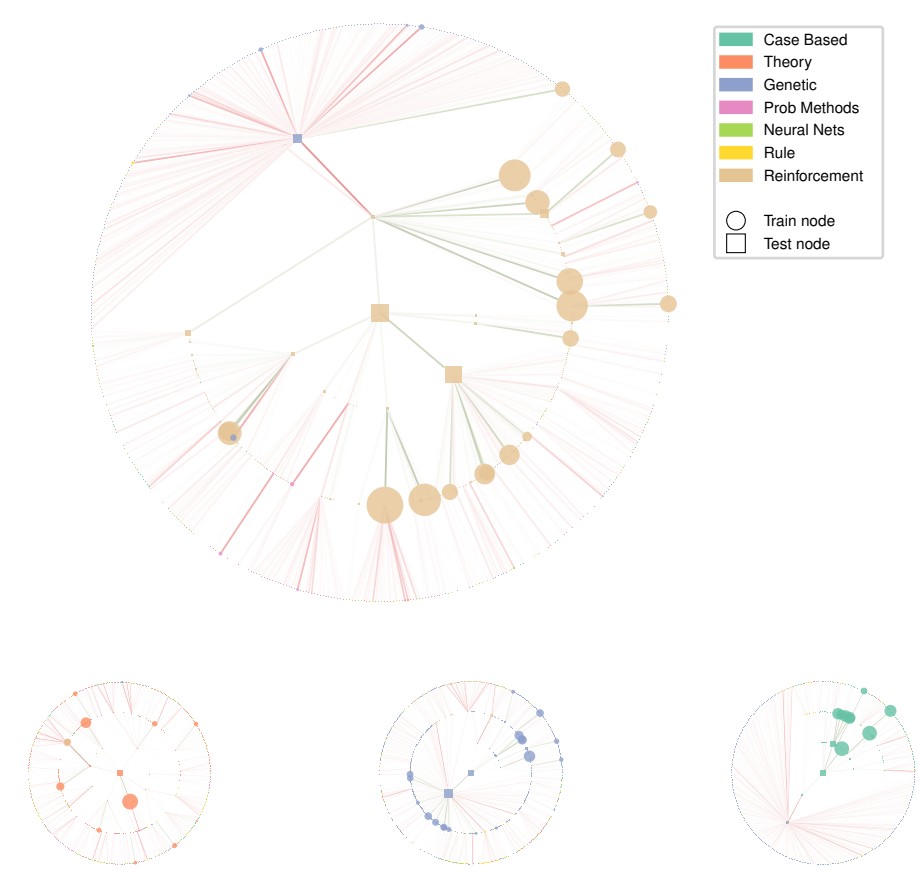

Figure 9: Importance of other nodes for a given test node according to the `learning` values of `DM`. Size of nodes represents their importance for the given one. The sign of the importance is given by the color of incoming edges (green for positive, red for negative).

as the utility metric for `CoraML` using $\alpha\text{-BANZ}$. We see almost identical performance. This suggests that the data values capture intrinsic properties of the data, rather than properties of the model. This also means that it is sufficient to use a cheaper model such as SGC to compute data values.

**Visualizations.** To gain insights into how predictions of nodes are influenced by others, we use the `DM` data values and look at the most positively (negatively) important nodes in the two-hop neighbors of selected test nodes. In Fig. 9, we can see the tree generated by a breadth-first search from a given test node, where colors represent classes, circles represent training nodes and squares non-training nodes. The size of each node is given by the absolute value of the importance of that node for the pre-

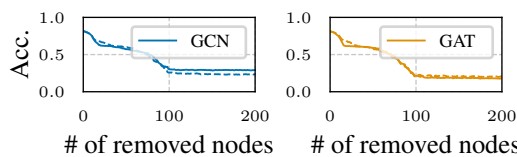

Figure 8: Transferability of SGC ranking for $\alpha\text{-BANZ}$ on `CoraML`. Dashed line is with transferred values.

diction of the root node. Edges are colored red if the node is negatively impacting the prediction, and green otherwise. The most important (positive) nodes are usually training nodes (big circles). However, there are also some important test nodes (big squares) which are more relevant than other training nodes. This emphasizes the importance of looking at both training and unlabeled nodes within the graph domain. Nodes from the same class (same color) have positive importance (green edges), while negatively important nodes come from other classes. See § F for more visualizations.

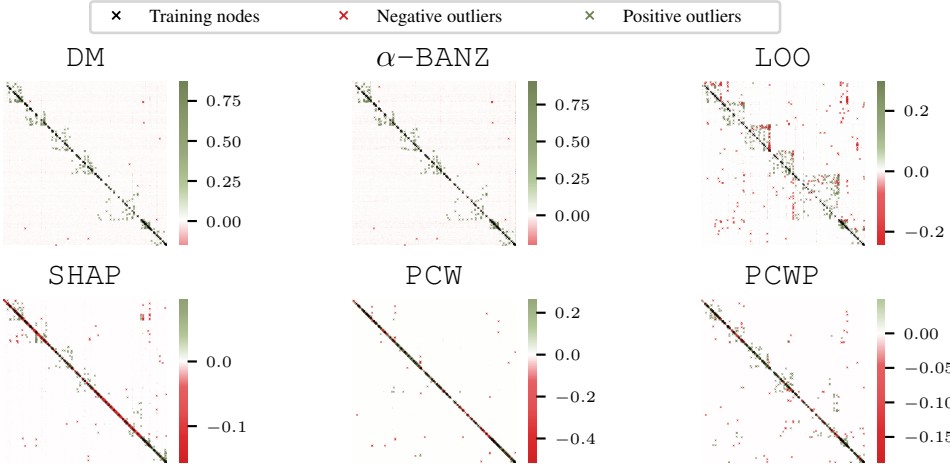

Figure 10: Heatmap of the `overall` signal values on `CoraML` with margins as utility. Green and red dots represent positive and negative importance respectively. Crosses represent highly impactful nodes (with values larger than the ones indicated in the bars). Black crosses mark the training nodes.

In Fig. 10, we show the heatmap of `overall` values with SGC as the model and margins as the utility. A visible pattern mostly for `DM` and $\alpha$-`BANZ` is the influence of nodes that belong to the same class which also agree with the results in Fig. 9. Another interesting observation is that columns of positive (green) values happen in correspondence with training node (black crosses), identifying training nodes that are important for many other nodes. Also, differently from other approaches `DM` and `BANZ` show less negative outliers (red crosses) that represent very negatively important nodes, compared to the positively important outliers (green crosses). For a more detailed analysis and visualizations of node values see § D.

## 5 RELATED WORKS

Data valuation has mainly been studied in the i.i.d. settings. Ghorbani & Zou (2019) was the first paper proposing to investigate how to attribute training datapoints generalizing Shapley values from features to datapoints. Similarly, Wang & Jia (2023) makes use of the Banzhaf values. More recent efforts investigate predictive approaches where surrogates are leveraged to approximate utility (Ilyas et al., 2022; Park et al., 2023; Wu et al., 2024). Approaches identifying input patterns to explain the model's prediction have been studied in the context of graphs as well (Duval & Malliaros, 2021; Akkas & Azad, 2024; Bui et al., 2024). They all use different semivalues at *inference* time to identify patterns in the input that explain the predictions. Finally, Chen et al. (2023); Chi et al. (2024) are the only two works accounting for data valuation in the context of graphs to relate input data to the model's performance. See § G for a more exhaustive discussion.

## 6 DISCUSSION AND CONCLUSION

In this paper, we present the first extensive study of data valuation methods for graph-structured data in a transductive semi-supervised setting. We introduce different data valuation scenarios, and apply state-of-the-art data valuation approaches that have not yet been investigated in the context of graphs. Our results demonstrate that these methods significantly outperform the latest efforts in graph data valuation across multiple applications. Moreover, we show how different utility functions can open up several applications. Due to the need to train numerous models on different subsets, the study we conducted was computationally intensive, and more effort is needed to develop more efficient data valuation methods for node values – both in terms of resources and time efficiency – in particular for very large graphs. Future directions also include exploring alternative proxies for datamodels that account for graph structures and a deeper investigation into settings where nodes are simply unlabeled rather than removed, as well as extending the analysis to the semi-supervised inductive setting.

## ETHICS STATEMENT

In this paper, we study data valuation for graphs. Data valuation is intended to study how data influence the training dynamics of neural networks making them more explainable. We don't see any particular ethical concern to mention about this study.

## REPRODUCIBILITY STATEMENT

To ensure the reproducibility of our results, we have provided a detailed explanation of the experimental setup and methodologies used in the main text in § 4. All models and hyperparameters are clearly described in § 4 and § E along with the adopted datasets, the sampling procedure and the model and specs of the machine used for the experiments. The code has been anonymized and submitted as part of the supplementary materials, available for download with guidelines on how to make it run.

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

## A    KEY TAKEAWAYS

In this section, we summarize the lessons learned and key insights from our study to guide future advancements in data valuation for graphs. Our findings contribute to understanding the strengths, limitations, and potential directions in this domain.

- **Context awareness:** Ad-hoc data valuation methods for graphs do not always perform better than state-of-the-art i.i.d. methods. Establishing a clear context and understanding domain-specific variations is critical before designing specialized approaches.

- **Brittle GNNs:** Data values are powerful tools for analyzing a model's behavior, such as the brittleness of GNNs to structural changes. For example, removing a few dozen nodes can lead to mispredictions for the majority of test nodes (§ 4). This complements adversarial analysis, opening opportunities for broader applications of data values. We believe data values can also contribute to exploring other related areas in a complementary manner.

- **Scalability:** Current data valuation approaches struggle with scalability due to the computational expense of a high amount of subset evaluations. Efforts should focus on developing methods that reduce this burden.

- **Maximal Sample Reuse (MSR):** Leveraging MSR allowed data Banzhaf to be more efficient in computing data values without compromising performance. Indeed, we empirically demonstrated that extending the latter from a uniform subset sampling to a binomial and adopting the MSR anyway, performs really well in practice while saving computation costs.

- **Simpler models for data valuation:** Data values are intrinsic to the data rather than specific models. Employing simpler, efficient models to compute data values (as supported by Fig. 8) can significantly lower computational costs.

These takeaways underscore the importance of efficient and context-aware approaches to data valuation for graphs. We hope our findings inspire further research into scalable, transferable, and computationally efficient methods that harness the full potential of data values.

## B    $\alpha$-BANZ WEIGHTING SCHEME

As discussed in § 2, most of the data valuation approaches can be referred to as semivalues and be defined by plugging in different weighting schemes in Eq. 2. We also have seen that data Banzhaf can be computed as $\phi_{\text{BANZ}}(i) = \mathbb{E}_{\mathcal{S} \sim \text{Unif}(2^{n \setminus i})}[u(\mathcal{S} \cup \{i\}) - u(\mathcal{S})]$ and this allows leveraging MSR for a more efficient approximation. Additionally, for datamodels, we sample subsets according to a binomial distribution controlled by the parameter $\alpha$, $\text{Binom}(n, \alpha)$, and if we set $\alpha = 0.5$ this equals data Banzhaf (Wang & Jia, 2023; Hammer & Holzman, 1992). For the property of semivalues weight function, it holds that $\sum_{k=1}^{n} \binom{n-1}{k-1} w(k) = n$ and for

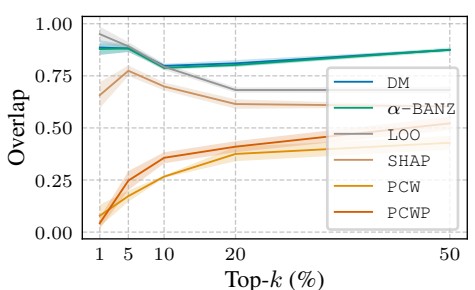

Figure 11: Rank overlap between `learning` and `overall` signal values computed with test margins as utility in the `all` setting.

data Banzhaf $w(k) = \frac{n}{2^{n-1}}$. As for $\alpha = 0.5$ datamodel equals data Banzhaf, we have that $w(k) = n \alpha^{k-1} (1 - \alpha)^{n-k}$.

We empirically show in § 4 that using the MSR trick for this new semivalue achieves competitive results compared to state-of-the-art data valuation approaches.

## C    EMPIRICAL LIMITATION OF GAME-THEORETIC APPROACHES

`SHAP` spends many evaluations for degenerate subsets containing very few nodes, as the value for a node is updated while progressively scanning each permutation. In contrast, approaches such as

DM and BANZ focus on subsets controlled by a probability parameter $\alpha$. This allows to evaluate subsets of more meaningful size, and a better assessment of a node's influence within some structural context. To address this, we slightly modified the data Shapley approach to consider more meaningful subsets. Specifically, during the permutation scan, subsets are independently augmented to include 10% of additional nodes (to align with $\alpha = 0.1$). This ensures that the value of the current node in the permutation is evaluated with a structural context influenced by other nodes in the subset. As in the original data Shapley, only the value of the current node in the permutation is updated. Fig. 12 verifies this showing that when the node's influence is evaluated with some structure context (dashed line), the computed Shapley values are more realistic. We leave it as future work exploring the optimal percentage of nodes to include for maximizing the effectiveness of SHAP.

## D   ANALYSIS OF NODE VALUES

This section analyses the assigned node values across different approaches and settings to determine the most efficient and accurate methods. Our analysis reveals that both DM and $\alpha$-BANZ consistently perform well across various settings. However, $\alpha$-BANZ offers a significant advantage by requiring less computational time, as it does not require learning a linear model for each node in $\mathcal{C}$, making it the more convenient choice. To perform this analysis, we focus on the values computed using SGC as a model on CoraML.

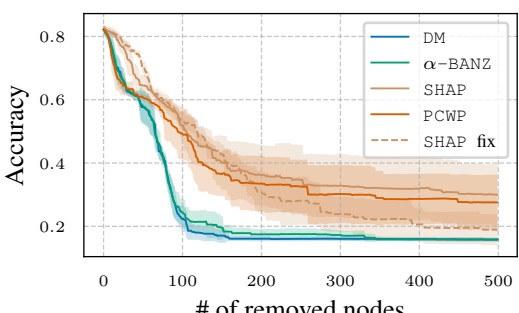

### D.1   LEARNING VS. OVERALL RANK

This experiment aims to explore the differences in rankings computed under the learning and overall settings. Fig. 11 illustrates the overlap in the top-$k\%$ of rankings generated by different approaches, with test margins as the utility

Figure 12: Most influential node removal with SGC as a model and test accuracy as the utility. SHAP fix indicates the improvement to account for degenerate subsets evaluations.

measure $u$, within the all setup. The results indicate that, regardless of whether a node is removed during both training and inference (overall signal), DM and $\alpha$-BANZ consistently recover rankings similar to those produced by the learning signal. This highlights the robustness of these methods in accurately predicting values across diverse settings, despite varying levels of available information, whereas other baselines exhibit significant changes in their rankings depending on the setup.

### D.2   RANKINGS OF APPROACHES

Fig. 13 shows the Kendall's $\tau$ coefficient between the ranking returned by the approaches for CoraML and SGC as the model. As also shown in the node influence experiment, DM and $\alpha$-BANZ are the most correlated approaches, and as expected PCWP is correlated with PCW.

Fig. 14 and Fig. 15 show the rankings of the approaches according to the learning signal using accuracy and margins as the utility respectively. Red bars correspond to training nodes, blue bars correspond to validation, and green bars correspond to test nodes. We can see that most training nodes are ranked as the most important even though we are in the all setting. However, some test nodes appear high in the rank, meaning they cause a decrease in the performance when removed from the graph. Once again, this confirms the importance of considering both the labeled and unlabeled nodes for attributing importance to the non-i.i.d. setting.

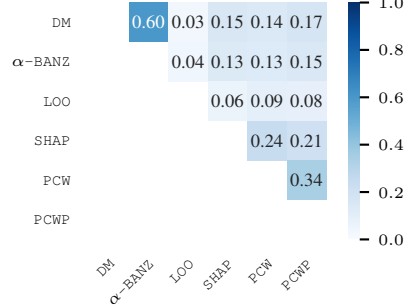

Figure 13: Kendall's $\tau$ between approaches.

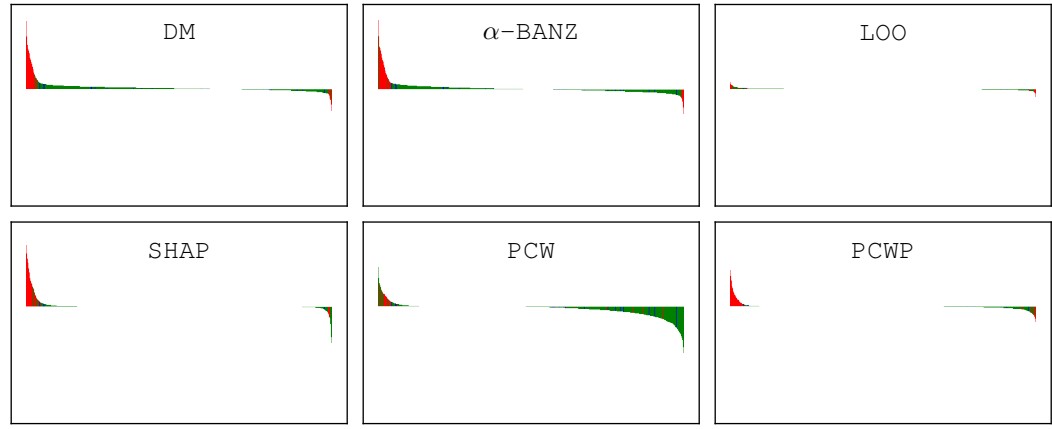

Figure 14: Ranking of nodes (training in red, validation in blue and test in green) according to the `learning` signal values considering test accuracy as the utility.

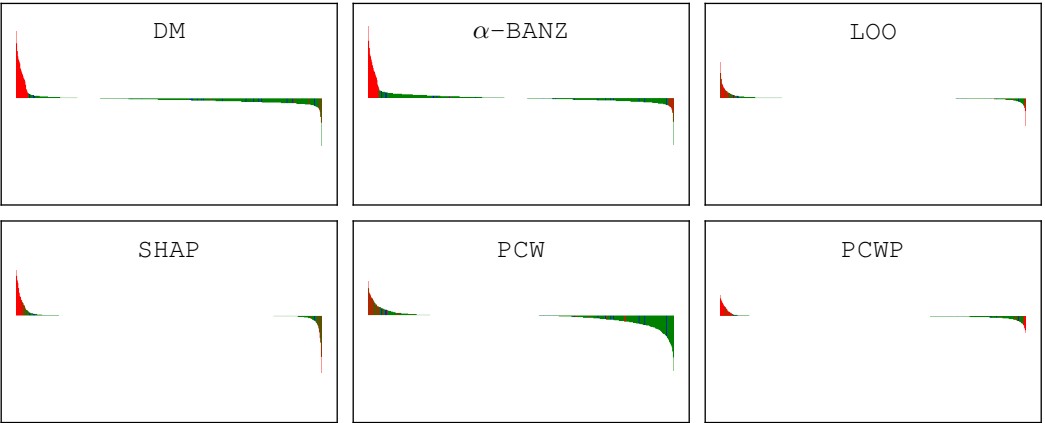

Figure 15: Ranking of nodes (training in red, validation in blue and test in green) according to the `learning` signal values considering test margins as the utility.

### D.3 NODE AND CLASS IMPORTANCE

**Node influence.** Similar to what we have presented in § 4, Fig. 16 shows the heatmap of the `learning` values of the approaches. The values are ordered according to the classes of the node. Similar considerations can be drawn as we can see positive (green) important nodes within clusters (same class). Furthermore, we see that where a training node is marked (black cross) we witness a vertical line (either green or red) suggesting that the training node influences the prediction of many others.

We show several bfs visualizations of the `overall` variant in Fig. 17, where we see how the value of the nodes are amplified but still mostly important nodes belong to the training (circles) nodes of the same class and some high-degree test node (squares).

**Cluster influence.** We can also see how nodes of a specific class influence on average the other classes. Fig. 18 and Fig. 19 show the `learning` and `overall` values respectively computed with SGC as a model. Interestingly, we can see how negatively important values fade when going from `learning` to `overall` in particular for the `PCW` and `PCWP` approaches. Additionally, we plot the confusion matrix of models' predictions over the influence scores heatmap to provide insights about in-class model certainty. We can see how in-class highly positive influence corresponds to the classes where the model predicts fewer false positives and thus is more certain about its predictions.

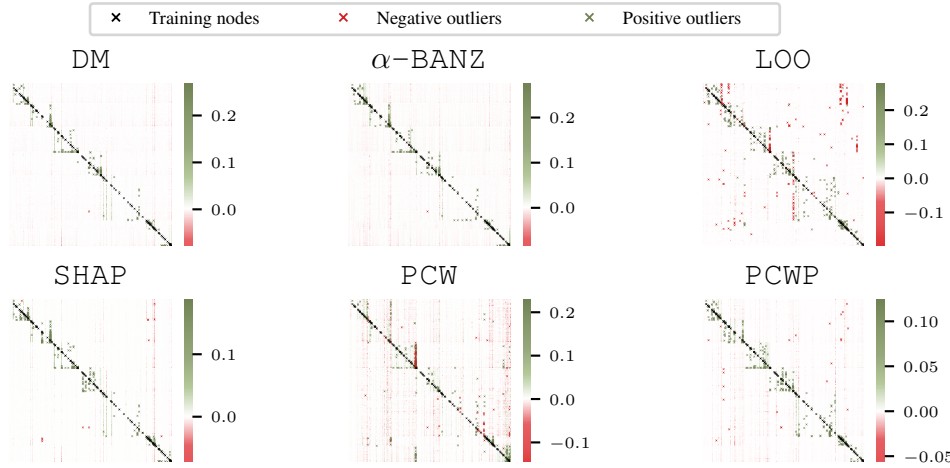

Figure 16: Heatmap of the `learning` signal values on `CoraML` with margins as utility. Green and red dots represent positive and negative importance respectively. Crosses represent highly impactful nodes (with values larger than the ones indicated in the bars). Black crosses mark the training nodes..

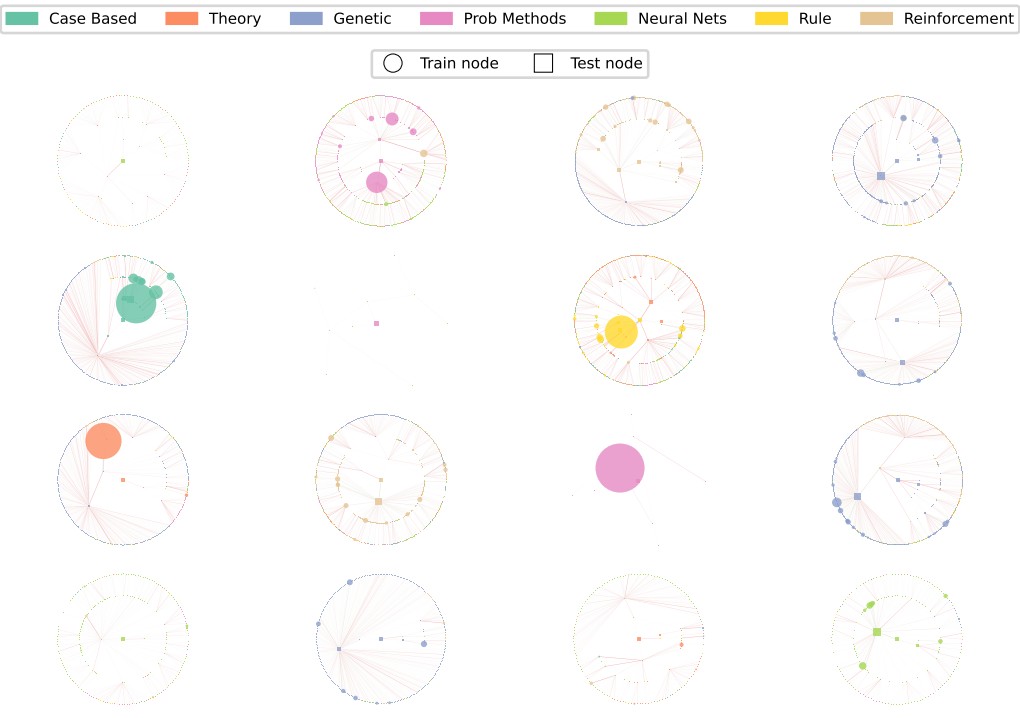

Figure 17: Importance of other nodes for test nodes according to the `overall` values of `DM`. Size of nodes represents their importance for the given one. The sign of the importance is given by the color of the incoming edge (green for positive, red for negative).

## E    FURTHER EXPERIMENT DETAILS

**Datasets and models.**    Table 1 summarizes the statistics of the datasets used for the experiments. According to (Shchur et al., 2018), in the semi-supervised transductive setting, we use a stratified sampling for selecting training nodes (20 nodes for each class) and have as many validation nodes

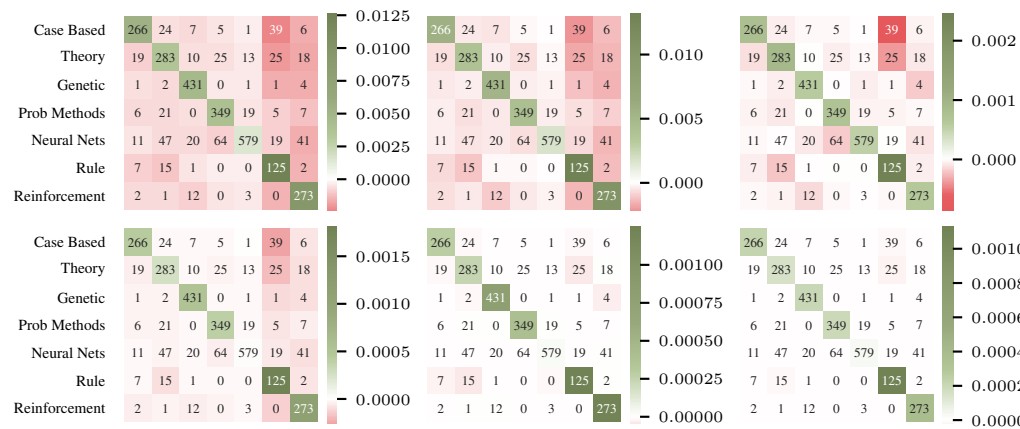

Figure 18: Average class-wise `learning` values with a SGC model. From left to right, the first row shows the results for `DM`, $\alpha$-`BANZ` and `LOO`. The second row is for `SHAP`, `PCW` and `PCWP`. Cell annotations are the confusion matrix values for model predictions.

Figure 19: Average class-wise `overall` values with a SGC model. From left to right, the first row shows the results for `DM`, $\alpha$-`BANZ` and `LOO`. The second row is for `SHAP`, `PCW` and `PCWP`. Cell annotations are the confusion matrix values for model predictions.

as the training. The remaining nodes constitute the test set. We select SGC, GCN and GAT as the architectures and fix the number of layers to 2 for all of them. For GCN and GAT, we select the number of hidden channels to be 16 along with a 0.5 dropout and ReLU activation function. For GAT, we adopt GATv2 version (Brody et al., 2022) with 8 attention heads.

Table 1: Summary of datasets.

| Dataset | Nodes | Edges | Features | Classes | Nodes (LCC) | Edges (LCC) | # Train/Val/Test |
|---|---|---|---|---|---|---|---|
| CoraML | 2 995 | 16 316 | 2 879 | 7 | 2 810 | 15 962 | 140/140/ 2 715 |
| Citeseer | 4 230 | 10 674 | 602 | 6 | 1 681 | 5 804 | 120/120/ 3 990 |
| PubMed | 19 717 | 88 648 | 500 | 3 | 19 717 | 88 648 | 60/ 60/19 597 |
| Photo | 7 650 | 238 162 | 745 | 8 | 7 487 | 238 086 | 160/160/ 7 330 |
| Computers | 13 752 | 491 722 | 767 | 10 | 13 381 | 491 556 | 200/200/13 352 |
| CoPhysiscs | 34 493 | 495 924 | 8 415 | 5 | 34 493 | 495 924 | 100/100/34 293 |

**Training setting.** We train GCN and GAT both using Adam optimizer with 0.01 as the learning rate and for 3000 epochs. An early stopping on the validation loss is set with 50 epochs as the

patience. We report average results on 10 different runs of each model (except for larger datasets where we train 5 models). Instead, for SGC we adopt a closed-form solution that is obtained by relaxing the classification problem to a regression one. Given the absence of non-linearities, the weights of the model can then be obtained as $W^\star = \tilde{X}_\ell H$ where $\tilde{X} = (\hat{X}^\top \hat{X} + \lambda I)^{-1} \hat{X}^\top$ with $\hat{X} = S^2 X$. Finally, for all the models, we run the experiments on 5 different train/val/test split and report an average of the performances. For training each node datamodel, we use the ridge regression with cross-validation implementation from scikit-learn (Pedregosa et al., 2011) with its default hyperparameters.

We found running experiments on the CPU to be faster than the GPU given the shallow architectures. We use the joblib library (Joblib Development Team, 2020) to parallelize the runs of different models and take advantage of the larger RAM availability than the GPU memory to run in parallel as many models as possible. We run the experiments on a cluster equipped with 136 nodes each with 2x AMD Epyc 9654 (96 Cores, 2.4-3.7 GHz) and 768GB RAM. For larger datasets, like `CoPhysiscs` we switch to a larger partition because of memory issues and use 8 nodes with the same specs but with 3TB RAM.

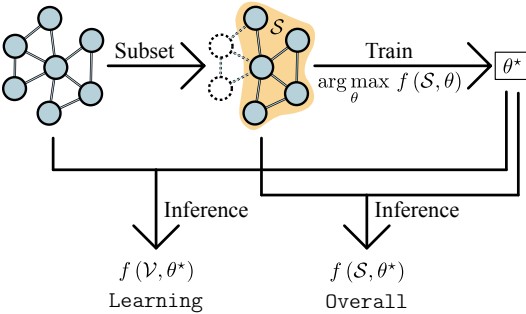

Figure 20: According to the graph in input to the trained GNN on $\mathcal{T}$, a different signal from the utility is measured – `learning` or `overall`.

**Running times.** The computational costs for calculating data values in an approach can be divided into two parts: the cost of collecting the actual utility across the subsets considered and the cost of computing the values. Table 2, Table 3, and Table 4 show the total of these two costs for each approach using 50 000 subsets with $\alpha = 0.1$. At parity of performances, $\alpha$-`BANZ` performs the best overall. '-' in the tables indicate costs that are yet to be collected.

Table 2: SGC computation times

| Dataset | DM | | $\alpha$-BANZ | | LOO | | SHAP | | PCWP | |
|---|---|---|---|---|---|---|---|---|---|---|
| | Train | All | Train | All | Train | All | Train | All | Train | All |
| Citeseer | 7 m 32 s | 11 m 38 s | 7 m 20 s | 5 m 7 s | 13 s | 29 s | 6 m 28 s | 3 m 24 s | 4 m 4 s | 3 m 57 s |
| CoraML | 41 m 30 s | 58 m 50 s | 41 m 13 s | 25 m 57 s | 25 s | 2 m 54 s | 43 m 16 s | 46 m 15 s | 40 m 35 s | 35 m 28 s |
| PubMed | - | - | - | - | 29 m 18 s | 2 d 6 h 57 m 7 s | 2 m 9 s | - | - | - |
| Photo | 7 h 53 m 49 s | 13 h 18 m 7 s | 7 h 46 m 41 s | 3 h 0 m 23 s | 2 m 4 s | 1 h 13 m 57 s | 8 h 25 m 21 s | 17 h 32 m 58 s | 7 h 41 m 21 s | 7 h 42 m 2 s |
| Computers | 1 d 20 h 18 m 23 s | - | - | 16 h 36 m 34 s | 19 m 8 s | 11 h 49 m 18 s | 1 m 8 s | - | - | - |
| CoPhysiscs | - | - | - | - | 3 h 8 m 27 s | - | 34 m 25 s | - | - | - |

Table 3: GCN computation times

| Dataset | DM | | $\alpha$-BANZ | | LOO | | SHAP | | PCWP | |
|---|---|---|---|---|---|---|---|---|---|---|
| | Train | All | Train | All | Train | All | Train | All | Train | All |
| Citeseer | 52 m 40 s | 16 m 33 s | 51 m 6 s | 9 m 46 s | 39 s | 2 m 22 s | 53 m 35 s | 28 m 2 s | 10 m 48 s | 10 m 48 s |
| CoraML | 3 h 53 m 11 s | 56 m 33 s | 3 h 54 m 16 s | 24 m 13 s | 1 m 7 s | 14 m 54 s | 3 h 48 m 52 s | 39 m 35 s | 17 m 39 s | 18 m 1 s |
| PubMed | 3 h 5 m 17 s | - | 3 h 2 m 40 s | - | 50 s | 1 h 28 m 26 s | 2 m 4 s | 2 h 20 m 5 s | - | - |
| Photo | 10 h 9 m 58 s | 14 h 24 m 27 s | 10 h 17 m 5 s | 34 m 50 s | 9 m 39 s | 5 h 22 m 32 s | 11 h 3 m 58 s | 1 h 44 m 26 s | 21 m 24 s | 21 m 25 s |
| Computers | - | - | - | - | 19 m 12 s | 18 h 5 m 51 s | 1 m 8 s | 5 h 22 m 29 s | 27 m 38 s | 27 m 34 s |
| CoPhysiscs | - | - | - | - | 17 m 23 s | - | 40 m 10 s | - | - | - |

Table 4: GAT computation times

| Dataset | DM | | $\alpha$-BANZ | | LOO | | SHAP | | PCWP | |
|---|---|---|---|---|---|---|---|---|---|---|
| | Train | All | Train | All | Train | All | Train | All | Train | All |
| Citeseer | 3 h 40 m 48 s | 25 m 53 s | 3 h 32 m 20 s | 19 m 36 s | 2 m 1 s | 10 m 22 s | 3 h 28 m 29 s | 1 h 0 m 43 s | 27 m 57 s | 27 m 57 s |
| CoraML | 15 h 15 m 20 s | 1 h 17 m 7 s | 15 h 2 m 52 s | 45 m 14 s | 4 m 56 s | 1 h 11 m 2 s | 16 h 50 m 56 s | 1 h 59 m 5 s | 44 m 45 s | 44 m 45 s |
| PubMed | 17 h 59 m 22 s | - | 17 h 46 m 60 s | - | 4 m 32 s | 10 h 56 m 16 s | 2 m 12 s | 9 h 58 m 24 s | - | - |
| Photo | - | 12 h 30 m 27 s | - | 1 h 17 m 51 s | 44 m 25 s | 1 d 5 h 18 m 20 s | 4 d 7 h 42 m 0 s | 7 h 9 m 39 s | 41 m 51 s | 41 m 33 s |
| Computers | - | - | - | - | 2 h 17 m 30 s | - | 1 m 3 s | 1 d 1 h 51 m 20 s | 1 h 14 m 26 s | 1 h 14 m 21 s |
| CoPhysiscs | - | - | - | - | 1 h 13 m 2 s | - | 38 m 14 s | - | - | - |

**Datamodels training.** Originally, `DM` for a target sample is trained by excluding subsets that include the target sample when it is part of the training set, to prevent information leakage (Ilyas et al., 2022)

(in particular when evaluating the predictive performance of the approach). However, this is not always applicable in the different graph settings – when computing the `learning` signal value of a node, we exclude subsets containing that node from the sampled subset distribution used to train the datamodel[3]. On the other hand, when computing the `overall` value, this exclusion is impractical as it leaves no subset available for training the datamodel.

### E.1  LEARNING VS. OVERALL SIGNAL

As explained in § 3, once the GNN is trained on the induced subgraph $\mathcal{T}$, we need to evaluate the utility function. In the graph domain, two possible ways arise which are illustrated in Fig. 20. In particular, after training our GNN (top), to obtain the utility we feed the trained GNN (hatched rectangle) either with the full graph adding back the removed nodes ($\mathcal{V}$), or we keep staying with the $\mathcal{T}$ (bottom). From the first case, as we add back the removed nodes, the utility will capture the signal of removing nodes only during learning – from this, the name `learning` signal. In the second scenario, the utility captures the signal when removing the nodes from both learning and inference steps – namely, the signal of the `overall` removal of nodes.

### E.2  NODE INFLUENCE - FROM MARGINS TO INFLUENCE SCORE.

In order to rank the nodes for the node influence experiment, we need a score for each node. However, when using margins as the utility, for each node we get the influence value of each other node for its prediction. To obtain the final score and then compute the ranking, we consider the average contribution of each node on the prediction of all the others. Practically, given the matrix of the influence scores, where the row $i$ contains the influence of the other nodes for its prediction, then we take the column-wise mean as the final score to use for ranking the nodes.

### E.3  OPTIMIZED PCW − PCWP

In `PCW` we only compute marginal contributions for nodes in the first $(1 - \mathrm{tr})$ portion of each node's child subtrees, approximating the marginal contributions of players in the remaining subtrees as $0$ and stop this procedure when the exact number of subsets $(50\,000)$ is reached. As this approach was developed for the inductive setting, we figured out that the selected truncation ratios were too tailored for the datasets used in that setting. It turns out that optimizing the truncation ratios to maximize the number of considered permutations leads to better performance for the approach in the transductive setting. For this reason, we set both the 1-hop and 2-hop neighbors truncation ratio to $0.99$ and we refer to this approach as `PCWP`.

### E.4  CHOICE OF THE HYPERPARAMETER $\alpha$

As mentioned in § 4, the parameter $\alpha$ represents the probability with which each node from $\mathcal{D}$ is kept in the subset or not. In other words, it establishes the size of the subsets. For instance, in the `train` setting where $\mathcal{D} = \mathcal{V}_t$, selecting $\alpha = 0.1$ results is subsets that are the $10\%$ of $\mathcal{V}_t$. This is a hyperparameter for the approaches `DM` and $\alpha$-`BANZ` and as such, it can be selected via a search. Given the time complexity of performing such a search for each dataset, we opted to select the best $\alpha$ for `CoraML` and adopt this across all the other datasets even though it could be suboptimal. However, as it is shown in § 4, the picked $\alpha$ is enough to outperform the other approaches for most of the datasets.

Fig. 21 shows how $\alpha$-`BANZ` performs in the most influential pruning experiment when subsets sampled with different $\alpha$s are provided to predict the node values. We can see that $\alpha = 0.1$ and $\alpha = 0.25$ are the best choices for realistic value estimates. However, as having fewer training samples results in faster training of the models, we decided to opt for $\alpha = 0.1$ as the best choice for $\alpha$.

---

[3]In the `all` setting, this subsets exclusion applies to $\mathcal{V}_u$ as well.

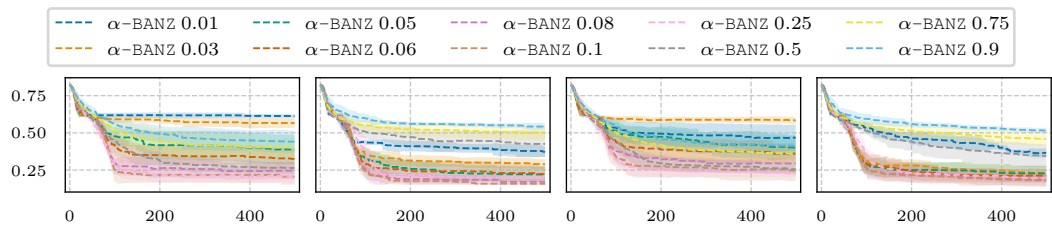

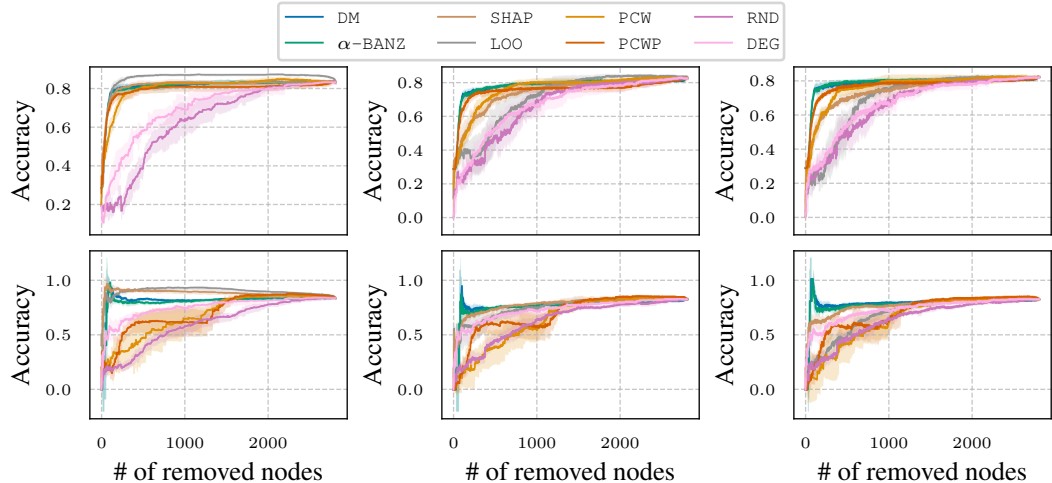

Figure 21: Hyperparameter search for the best $\alpha$ for sampling the subsets employed in the node value predictions for the $\alpha$-BANZ approach. The test accuracy and the number of removed nodes are shown in the vertical and horizontal axes respectively. From left to right, the figure reports the results for `learning` & test margins utility, `learning` & test accuracy utility, `overall` & test margins utility and `overall` test accuracy utility.

Figure 22: Most influential node addition for different models – SGC (first column), GCN (second column) and GAT (third column) – according to the `learning` (first row) and `overall` signal values (second row).

## F  ADDITIONAL EXPERIMENTS

### F.1  MOST INFLUENTIAL ADDITION

In Fig. 22, we show the results for the most influential node addition experiment. We present results computed according to both `learning` and `overall` values. Contrary to the most influential pruning, here we expect a steep increase in the model's performance while adding the most influential nodes, followed by a plateau. As expected, also here DM and $\alpha$-BANZ better assess node values.

### F.2  NODE INFLUENCE IN THE TRAIN SETTING

Here we present the results for the node influence experiment in the `train` setting, namely we attribute importance (and then remove) only training nodes. In Fig. 23 we show how the approaches perform considering both the `learning` and `overall` signals, across different models (SGC, GCN and GAT). We can see how the approaches struggle more in assessing the right values when non-linearities are introduced in the models (GCN and GAT). In any case, DM and $\alpha$-BANZ perform better than the others.

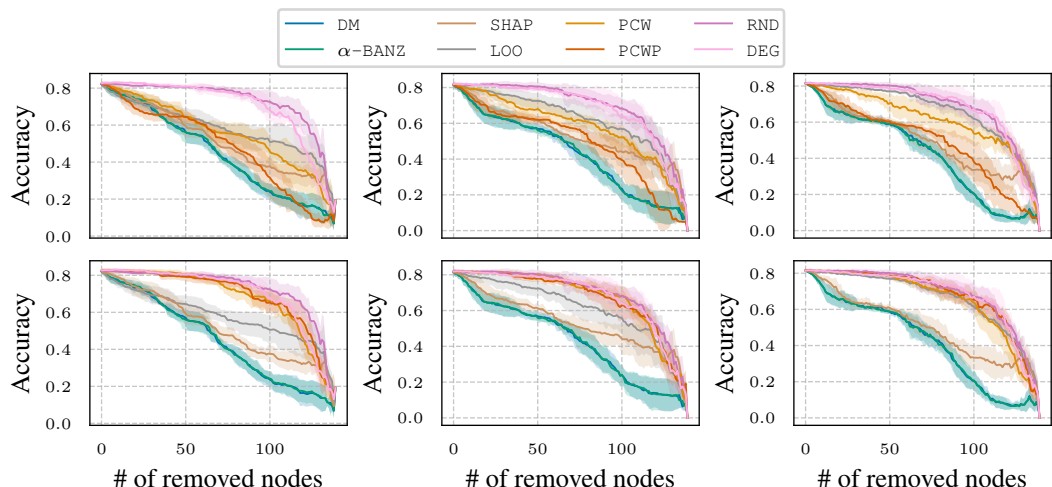

Figure 23: Most influential node pruning for different models – SGC (first column), GCN (second column) and GAT (third column) – according to the `learning` (first row) and `overall` signal values (second row).

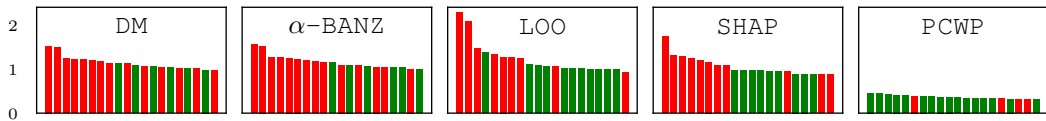

Figure 24: Rank of training nodes according to values computed on a graph with 10% of poisoned nodes. Red and green bars represent poisoned and not-poisoned nodes respectively.

### F.3 MEMORIZATION RANKING

Fig. 24 shows the rankings of the approaches as a result of the memorization experiment. As explained in § 4, we poison 10% of the training data and expect a good data validation approach to rank these nodes as the most important for their prediction. We show the results for `CoraML`, where the number of training nodes is 140 resulting in 14 poisoned nodes. The results show that `DM` and $\alpha$-`BANZ` rank first the poisoned nodes establishing as robust approaches for detecting poisoned or mislabeled data.

### F.4 STABILITY

One important aspect of a reliable data valuation approach is its consistency producing similar ranking results across multiple runs of the same experiment. Fig. 25 illustrates the stability of the different approaches, measured across 10 different seeds for each of the considered 5 train/val/test splits in `CoraML`. Each of the 10 different seeds will then define different subsets/permutations to scan for assessing node values. We report the average performance of each method and observe that `DM` and `BANZ` exhibit the lowest variance across runs (on the left-hand side of the figure), making them the most reliable approaches for data valuation on graphs.

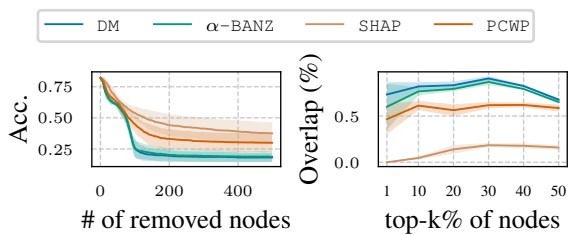

Figure 25: Approaches stability: (left) in predicting node values; (right) in predicting the ranking for a given split across 10 seeds.

We also report the ranking stability (right-hand side of Fig. 25), where we measure the overlapping top-$k$ percentage of nodes of the first $300$ nodes in the `CoraML` node ranking. Also here, we can see that approaches like `DM` and `BANZ` mostly predict the same rankings across different seeds for a given train/val/test split.

## G    MORE RELATED WORKS

In the literature, data valuation has been adopted mainly in an i.i.d. setting to study relations between training data and changes in the model's utility or to support prediction's explanations at inference time. This section presents an overview of the works that are closest to ours.

**Graph data valuation at inference time.**    When applied at inference time, a data valuation approach assesses the most important input patterns that mostly influence the final model's prediction. Particularly for graphs, this translates into highlighting the graph structure or node features that cause the prediction for the input. For instance, Duval & Malliaros (2021) compute graph structure and node feature explanations for a single example by constructing a surrogate model on a perturbed graph and computing Shapley values as explanations. Akkas & Azad (2024) explain predictions by computing Shapley values edge-wise and outputting the subgraph with edges from the top-k values. Chhablani et al. (2024) use Banzhaf value in combination with thresholded utility functions on edges to provide counterfactual explanations. Finally, Bui et al. (2024) introduce the Myerson-Taylor index which includes structure information inside the Shapley value to identify important motifs for the prediction. Differently from the mentioned works, we employ data valuation to discover relations between nodes involved in the training and the final output of the model. Namely, we do not treat the GNN as a fixed black box function.

**Data valuation in the i.i.d. setting.**    The seminal work by Ghorbani & Zou (2019) extends Shapley value from feature-level to data point granularity to assess training data importance. As this may result in a computationally expensive procedure, Jia et al. (2021) compare the utility of different data attribution methods and propose a fast estimator for Shapley values based on k-nearest neighbors surrogate. A more recent research direction uses linear surrogates to learn a mapping from a subset of the training set to the model's utility (Ilyas et al., 2022). The surrogate is then used to estimate the utility from newly sampled subsets and the surrogate weights represent the data values. As the number of possible training subsets is exponential w.r.t. the number of training data points, the learning of the surrogate may be time-consuming. In a follow-up work, Park et al. (2023) employ Taylor approximation to linearize a considered deep neural network such that the one-step Newton approximation can be applied in a non-linear setting. They show that in such a way, just a few training subsets are needed to attribute importance to training data accurately. Ultimately, Wang & Jia (2023) introduce the concept of data Banzhaf for assessing the values of training data and showing its robustness in differentiating data. Except for data Shapley, these approaches have not yet been applied to the graph domain. In our work, we incorporate and compare all these approaches to investigate their behaviour when structural information comes into play.

**Data valuation for graph-structured data.**    Graph data valuation relating data to model predictions is still under investigation. As far as we are concerned, only two works try to tackle this direction. Chen et al. (2023) adapt influence functions to a graph-structured model to approximate the leave-one-out training. However, they consider only training nodes in a transductive setting, ignoring in such a way the interactions between unlabeled and labeled nodes. Differently from them, we consider a more comprehensive picture by studying the transductive scenario, looking at the influence of all nodes, whether labeled or not. Instead, Chi et al. (2024) propose a new coalition sampling based on the Winter value that considers the graph's structure when creating a permutation of nodes to process and estimate each node's contribution. However, this work focuses on the inductive scenario where unlabeled nodes do not play a role during the model training.

