# OpenReview forum: "Data Valuation for Graphs"
_ICLR.cc/2025/Conference — Submitted to ICLR 2025_

### Official Review · Reviewer_uvDP · 2024-11-03

**Soundness:** 3
**Presentation:** 3
**Contribution:** 2
**Rating:** 3
**Confidence:** 3

**Summary:**

This is a benchmark paper on data valuation for graphs. It summarizes and compares different data value notions, whether to predict the estimation of data value, and different sampling methods.

**Strengths:**

- The summarization and survey of previous data valuation methods are clear and detailed.

- The experiments and evaluation are sufficient and intriguing.

**Weaknesses:**

- The analysis of the experiment results does not match the illustrated figures. See questions.

- No running time is provided. I'm concerned about how the sampling subset number affects the value estimation and running time, considering that 50,000 subsets is quite a large number, even for small graphs.

**Questions:**

- What's the main challenge of applying game-theoretic data notions to graphs? Do we need any specific designs for the graph besides customizing the influence function?

- In Fig. 3, it seems that in subgraphs 5&6, DM does not outperform other methods. Can you clarify that?

- It's interesting to control the average size of coalitions. What about controlling the upper bound of size, e.g., to 500, since in experiments, we only remove the top 500 nodes?

- Did you make any adaptations from Chi's PC-Winter to the transductive setting? How do you exclude the effect by the test node that has already been seen during training?

---

> ### Author Response · Authors · 2024-11-20
>
> We would like to thank the reviewer for the detailed comments. We appreciate you find our categorization clear and detailed, and the experiments varied. In the following we response to the comments.
>
> **W2)** The cumulative runtime for all the experiments is already mentioned in the introduction. The runtime of each approach can be divided into two times: one needed to collect the utility across the subsets, and one needed to compute the values for the nodes. In the following table we report the sum of these two times:
>
> **SGC Computation times**
>
> | Dataset | DM `Train` | DM `All` | α-BANZ `Train` | α-BANZ `All` | LOO `Train` | LOO `All` | SHAP `Train` | SHAP `All` | PCWP `Train` | PCWP `All` |
> | --- | --- | --- | --- | --- | --- | --- | --- | --- | --- | --- |
> | Citeseer | 7 m 32 s | 11 m 38 s | 7 m 20 s | 5 m 7 s | 13 s | 29 s | 6 m 28 s | 3 m 24 s | 4 m 4 s | 3 m 57 s |
> | CoraML | 41 m 30 s | 58 m 50 s | 41 m 13 s | 25 m 57 s | 25 s | 2 m 54 s | 43 m 16 s | 46 m 15 s | 40 m 35 s | 35 m 28 s |
> | PubMed | - | - | - | - | 29 m 18 s | 2 d 6 h 57 m 7 s | 2 m 9 s | - | - | - |
> | Photo | 7 h 53 m 49 s | 13 h 18 m 7 s | 7 h 46 m 41 s | 3 h 0 m 23 s | 2 m 4 s | 1 h 13 m 57 s | 8 h 25 m 21 s | 17 h 32 m 58 s | 7 h 41 m 21 s | 7 h 42 m 2 s |
> | Computers | 1 d 20 h 18 m 23 s | - | - | 16 h 36 m 34 s | 19 m 8 s | 11 h 49 m 18 s | 1 m 8 s | - | - | - |
> | CoPhysics | - | - | - | - | 3 h 8 m 27 s | - | 34 m 25 s | - | - | - |
>
> ---
>
> **GCN Computation times**
>
> | Dataset | DM `Train` | DM `All` | α-BANZ `Train` | α-BANZ `All` | LOO `Train` | LOO `All` | SHAP `Train` | SHAP `All` | PCWP `Train` | PCWP `All` |
> | --- | --- | --- | --- | --- | --- | --- | --- | --- | --- | --- |
> | Citeseer | 52 m 40 s | 16 m 33 s | 51 m 6 s | 9 m 46 s | 39 s | 2 m 22 s | 53 m 35 s | 28 m 2 s | 10 m 48 s | 10 m 48 s |
> | CoraML | 3 h 53 m 11 s | 56 m 33 s | 3 h 54 m 16 s | 24 m 13 s | 1 m 7 s | 14 m 54 s | 3 h 48 m 52 s | 39 m 35 s | 17 m 39 s | 18 m 1 s |
> | PubMed | 3 h 5 m 17 s | - | 3 h 2 m 40 s | - | 50 s | 1 h 28 m 26 s | 2 m 4 s | 2 h 20 m 5 s | - | - |
> | Photo | 10 h 9 m 58 s | 14 h 24 m 27 s | 10 h 17 m 5 s | 34 m 50 s | 9 m 39 s | 5 h 22 m 32 s | 11 h 3 m 58 s | 1 h 44 m 26 s | 21 m 24 s | 21 m 25 s |
> | Computers | - | - | - | - | 19 m 12 s | 18 h 5 m 51 s | 1 m 8 s | 5 h 22 m 29 s | 27 m 38 s | 27 m 34 s |
> | CoPhysics | - | - | - | - | 17 m 23 s | - | 40 m 10 s | - | - | - |
>
> ---
>
> **GAT Computation times**
>
> | Dataset | DM `Train` | DM `All` | α-BANZ `Train` | α-BANZ `All` | LOO `Train` | LOO `All` | SHAP `Train` | SHAP `All` | PCWP `Train` |  |
> | --- | --- | --- | --- | --- | --- | --- | --- | --- | --- | --- |
> | Citeseer | 3 h 40 m 48 s | 25 m 53 s | 3 h 32 m 20 s | 19 m 36 s | 2 m 1 s | 10 m 22 s | 3 h 28 m 29 s | 1 h 0 m 43 s | 27 m 57 s | 27 m 57 s |
> | CoraML | 15 h 15 m 20 s | 1 h 17 m 7 s | 15 h 2 m 52 s | 45 m 14 s | 4 m 56 s | 1 h 11 m 2 s | 16 h 50 m 56 s | 1 h 59 m 5 s | 44 m 45 s | 44 m 45 s |
> | PubMed | 17 h 59 m 22 s | - | 17 h 46 m 60 s | - | 4 m 32 s | 10 h 56 m 16 s | 2 m 12 s | 9 h 58 m 24 s | - | - |
> | Photo | - | 12 h 30 m 27 s | - | 1 h 17 m 51 s | 44 m 25 s | 1 d 5 h 18 m 20 s | 4 d 7 h 42 m 0 s | 7 h 9 m 39 s | 41 m 51 s | 41 m 33 s |
> | Computers | - | - | - | - | 2 h 17 m 30 s | - | 1 m 3 s | 1 d 1 h 51 m 20 s | 1 h 14 m 26 s | 1 h 14 m 21 s |
> | CoPhysics | - | - | - | - | 1 h 13 m 2 s | - | 38 m 14 s | - | - | - |
>
> Experiments are still running and we’ll update the tables as they finish. We added the tables in the manuscript (see Appendix D).
>
> **W1/Q1)** As we explained in section 3, when we need to assess the value in the non-i.i.d. setting we have different scenarios that we need to take into account according to the downstream application. First of all, as also unlabeled nodes play a role in the prediction for others (via their structure), accounting only for the value of training examples (as done in the i.i.d. setting) might results in suboptimal attribution. We show this in our experiments, for example in Figure 9, we have a test node that is more influent than other training nodes for the prediction of a given node (this is also shown in Figures 13-14 where some test nodes is ranked earlier than some training node). Another underestimated aspect is the way the utility is computed: once we train the model on a subset, we can compute the utility either on the full graph (adding back the removed nodes) — that we address as `learning` signal — or keeping the subgraph — that we address as `overall` signal. We also added a new figure (Figure 19) that makes clear the distinction of these two settings. More generally, coming up with new graph-specific approaches might be tricky if we do not properly account for the baselines in the right setting. Indeed, as we show in our experiments, already state-of-the-art approaches for the i.i.d setting perform really well in the semi-supervised learning.

---

> > ### Author Response · Authors · 2024-11-20
> >
> > **W1\Q2)** As we mention in line 328-329 (line 331 of the revised version), in Figure 3 subgraphs 3&5&6 DM is omitted because its computation causes out-of-memory. We are currently working on more memory efficient implementation and we will post results if they are finished before the rebuttal deadline.
> >
> > **W1/Q3)** When controlling the coalition size via $\alpha$ we are implicitly controlling the upper bound as well, because of the property of the binomial. For example, in our experiments with $\alpha$ set to $0.1$ for `CoraML` dataset we have an average subset size of $280.98$, that correspond to the $10\%$ of the nodes in the graph. For $\alpha = 0.1$ we can explicitly calculate the probability of sampling a subset with $500$ or more nodes, the probability is essentially $0$. Indeed, out of our $50000$ subsets the maximum size is 348. From the node pruning experiment, we only remove the first $500$ nodes because we are interested the most in the effect of removing the high-valued nodes. Indeed, we can see that already after removing a few hundred nodes we reach a plateau of very low performance indicating that from there meaningless nodes are being removed.
> >
> > **W1/Q4)** The only adaptation we did to PCW is to make it work for the transductive setup — all the test nodes are available both at training and test time. We keep the actual computation of the PCW as it has been proposed. We moreover, include a different variant aiming at maximizing the number of permutations to scan via increasing the ratio of neighbor nodes. We show that this variant performs better for the transductive setting.

---

> > > ### Comment · Reviewer_uvDP · 2024-11-25
> > >
> > > Thank you for rebuttals. I think the authors have done enough amount of implementation work, while their novel contribution is limited. The data valuation seems still time-consuming and greatly limits its application to graphs. Thus, I keep my score.

---

### Official Review · Reviewer_FLZg · 2024-11-04

**Soundness:** 3
**Presentation:** 2
**Contribution:** 2
**Rating:** 5
**Confidence:** 5

**Summary:**

This paper presents a comprehensive study of existing data valuation techniques on graphs. It finds that datamodel and data Banzhaf outperforms other method (including graph-specific valuation). The authors also show the potential of using node values for various applications.

**Strengths:**

S1. This paper offers a thorough introduction to data valuation and valuation for graphs.

S2. It evaluates various valuation techniques on semi-supervised node classification in several different downstream tasks.

**Weaknesses:**

W1. It is unclear whether removing nodes from a graph can be considered as a valuation for node itself, given that message passing depends on edges as well. Node classification accuracy can change even if keeping the node itself and only remove edge(s) from the graph (e.g., how poisoning attacks are done on graphs). Essentially I think these valuation is measuring the values of a node and its associated edges, rather than just the value of **a node**.

W2. As a paper that focuses on comprehensive study of existing methods, I would like to see more analyses on why some methods offer good valuation while some don't, especially for PCW which is the only one method that designed for graphs. The current section 4 is more like describing all results, without many insightful analyses or discussion to dig deeper into the results, which should be the key but missing contribution of this paper to the field.

W3. There are quite a lot of figures and heatmaps in the paper. I would suggest the authors to add necessary legends and figure captions into the paper for better clarity. It is also unclear why different circles have a very different sizes in Figure 9.

W4. I think the paper could be further strengthened if it includes a dedicated section to discussion the lessons learned and what actions the community should take for precise valuation for graphs.

**Questions:**

Please see weaknesses.

---

> ### Author Response · Authors · 2024-11-20
>
> We would like to thank the reviewer for the detailed comments. We appreciate you find our work a good introduction to the field, and the evaluation of the approaches in downstream tasks varied. In the following we response to the comments.
>
> **W1)** Thank you for pointing out this aspect. It is not clear to us what does it mean to remove a node but keep its edges. We are open to suggestions from the review. Nonetheless, we agree that is more clear to say that value we are computing is the value of the node and its associated edges. We updated the paper to reflect this (see lines 321-323).
>
> **W2)** Our work focuses more in establishing a proper setting for data valuation in the graph domain. As a consequence of this, we believe that before developing ad-hoc strategies for the domain under investigation we should properly define the problem and correctly evaluate the baselines. Section 4 provides general insights about the functioning of graph neural networks and graph data rather than analysing already established approaches. For example, we pointed out how state-of-the-art approaches for the i.i.d. setting already perform better than ad-hoc graph methods (see Figure 3 and relative section). We showed the brittleness of GNNs where when just removing a few tens of nodes results in the model to mispredict most of the test nodes (see figure 4[left] and relative paragraph. We showed how baseline approaches are better in detecting poisoned nodes in the dataset than ad-hoc graph methods (see Figure 7 and relative paragraph). We also show how data values can transfer across models (Figure 8) and how values can be used to study how predictions are affected by the other nodes (see Figure 9 and 10).
>
> An insight on why PCW fails in capturing meaningful values might be its similarity to data Shapley. Specifically, PCW restrict the possible permutations of Shapley such that it accounts for the graph structure ($k$-hop neighbors), and then scan these permutations as done in data Shapley. Then, as for data Shapley many evaluations are performed on degenerate subsets made of a very few nodes (each permutation is scanned in a progressive way). Instead, considering subsets controlled by a probability $\alpha$ of keeping a node — as for Datamodel and data Banzhaf, allows to use most of the evaluations for subsets of meaningful size. Another reason might be PCW’s inductive nature, as it was designed for inductive tasks. Indeed, we showed that in the transductive setting, maximixing the permutations (PCWP by increasing the truncation ratio for the 1- and 2-hop neighbors) benefits the final attribution. To verify this, we also implement the Banz and DM inductive variant to test the unlabeled node removal performance as in PCW. Preliminary results show that PCW in the inductive setting is better than DM and Banz.
>
> **W3)** The sizes of the nodes in Figure 9 are according to the magnitude of its importance (larger nodes are more important than smaller). We explained this in the text when introducing the figure. However, we agree with the reviewer that a legend can simplify the understanding of the figure. We have included a legend in Figure 10 too, where (although already mentioned in the caption) explaining the meaning of the crosses. The value for the color is implicitly given by the color bar. Could you please mention other figures which should be clarified more?
>
> **W4)** We agree with the reviewer that a dedicated section with the lesson learned and potential guidelines for the community is beneficial for the paper and we added it in the new revision (see Appendix A). We can integrate this section in the main body for the camera-ready version. Overall, the main insights can be summarized as:
>
> 1. It is not given that ad-hoc data valuation approaches for graphs work better than the current state-of-the-art in the i.i.d. setting. As we show, setting up and tuning the baselines indeed lead to better performance.
> 2. The Maximal Sample Reuse (MSR) adopted by data Banzhaf is a useful technique that provides better estimation because of the sample reuse. Moreover, we showed that this can be adopted with a binomial probability rather than uniform sampling and this empirically performs well.
> 3. GNNs are very brittle to structure modification — the removal of just a handful of nodes causes the model to mispredict most of the test nodes — and we showed this from a data attribution point of view, that is complimentary to adversarial perturbations.
> 4. Data values align more closely with true values when we can access a large number of subsets. However, this comes with increased computational costs. Developing more time-efficient methods is crucial for achieving more reliable data attributions. One potential approach is to leverage more efficient models, as data values are an intrinsic property of the data rather than the specific model used to compute them.

---

> > ### Comment · Reviewer_FLZg · 2024-11-21
> >
> > Thank you for your efforts in addressing my concerns. Most of my concerns are addressed, except for W2 and a minor thing about W3.
> >
> > W1: I think there is no way to remove a node and keep edges. A might-be-relevant setting could be more like treating a labeled training node as an unlabeled node, so you don't backprop any info of that node. But that might not be a 'real' valuation of a node since it is still present in the graph. I believe that making the change of node + associated edges is a more precise wording here.
> >
> > W2: I think my concern is more about offering more analysis to show possible reasons why some methods would fail. I think the second paragraph you offered is great, but it would be further strengthened with some results to support your claims. I think this is equally important to establishing a setting, i.e., (1) what the new setting is (which is what you are doing now) and (2) what the limitation of methods developed under old settings are (which is something I am interested). I think this is a way to further strengthen the paper. I feel the current experiments are a bit shallow in understanding the limitations of existing methods.
> >
> > W3: Thank you for your explanation. One more concern I have is that several baseline methods have very similar coloring. For example, in Figure 6, SHAP, PCW, and PCWP have very similar colors if we don't zoom in to check carefully. It might be better to consider changing colors for these methods to avoid any confusions.
> >
> > W4: Thank you for adding this new section.

---

> > > ### Author Response · Authors · 2024-11-23
> > >
> > > Thank you for your comments. We appreciate you find most of your concerns addressed. In the following, we address the remaining points.
> > >
> > > **W2)** We appreciate your suggestion and agree that this analysis strengthens our work. To address your point, we slightly modified the data Shapley approach to account for evaluations of degenerate subsets. Specifically, while scanning a permutation, instead of only including nodes up to the current one, we also randomly include 10% of the other nodes. This ensures that the initial subsets contain more structure, providing a better context to evaluate the current node's influence. Figure 12 in the revised version (appendix C) demonstrates that, with this fix, the values computed by SHAP are even better than those of PCW.
> > >
> > > Regarding PCW, we discovered a possible data leakage in the original code (which we don't have in our pipeline). We are currently testing this more thoroughly to confirm.
> > >
> > > **W3)** Thank you for pointing this out. The initial coloring was intended to reflect similarities in the way the values are computed by the different methods. However, we agree that this may lead to confusion. We will update the color scheme in the next revision.

---

> > > > ### Comment · Reviewer_FLZg · 2024-11-24
> > > >
> > > > Thank you for the further response. I have increased my score. It would be great if the authors could address the data leakage in PCW and update the results as well.

---

### Official Review · Reviewer_5Tw8 · 2024-11-04

**Soundness:** 3
**Presentation:** 3
**Contribution:** 2
**Rating:** 5
**Confidence:** 4

**Summary:**

This work conducts a benchmark study of data valuation methods on graph neural networks. This work does not introduce new data attribution methods but focus on evaluation. The methods being evaluated include several generic data valuation methods (leave-one-out, Data Shapley, Data Banzhaf and Datamodels), as well as recent methods tailored for graphs. The results show that Data Banzhaf and Datamodels perform the best.

**Strengths:**

This paper conducts a comprehensive evaluation of existing data valuation methods for graph neural networks. This includes a reasonably large set of methods, and a variety of evaluation metrics. The visualization of influential nodes also provides helpful insights.

**Weaknesses:**

1. Overall, this paper lacks technical novelty as it only focuses on evaluating existing methods.

2. A major contribution claimed by this paper is the application of data valuation methods to the graph setting. However, most of the generic data valuation methods studied in this paper, such as leave-one-out, Data Shapley or Data Banzhaf, are not restricted to i.i.d. settings by design and can be straightforwardly applied to graph neural networks with proper choice of utility function. So the incremental novelty of generalizing these methods from a feedforward NN to a GNN is no significantly larger that that from a CNN to a ResNet.

3. The computation cost for the evaluated methods is not reported, which is a very important metric.

4. The experimental setups are simple GNNs on what have been considered as toy datasets by the GNN community.

**Questions:**

See Weaknesses.

---

> ### Author Response · Authors · 2024-11-20
>
> We would like to thank the reviewer for the detailed comments. We appreciate you find our evaluation comprehensive and the visualizations insightful. Please find below our response to the comments.
>
> **W1, W2)** As we mentioned in the introduction, our focus is not on the technical novelty. Rather, our focus is more on testing the state-of-the-art data valuation approaches in the context of graphs. Indeed, as we show ad-hoc graph approaches (PCW) are not necessarily better when compared to these baselines in the transductive setting. In addition, we introduce the concepts of `Learning` vs `Overall` values which uniquely applies to semi-supervised setting and has not been studied so far.
>
> **W3)** We already reported in the introduction the overall computation costs of running all the experiments included in the manuscript. However, computation costs of an approach can be split into two costs: one regarding the collection of the actual utility across the considered subsets, and then the cost of computing the value. In the following table we report the sums of these two costs for each approach with $50000$ subsets $\alpha = 0.1$. Overall, Banz is faster than DM across all settings.
>
> **SGC Computation times**
>
> | Dataset | DM `Train` | DM `All` | α-BANZ `Train` | α-BANZ `All` | LOO `Train` | LOO `All` | SHAP `Train` | SHAP `All` | PCWP `Train` | PCWP `All` |
> | --- | --- | --- | --- | --- | --- | --- | --- | --- | --- | --- |
> | Citeseer | 7 m 32 s | 11 m 38 s | 7 m 20 s | 5 m 7 s | 13 s | 29 s | 6 m 28 s | 3 m 24 s | 4 m 4 s | 3 m 57 s |
> | CoraML | 41 m 30 s | 58 m 50 s | 41 m 13 s | 25 m 57 s | 25 s | 2 m 54 s | 43 m 16 s | 46 m 15 s | 40 m 35 s | 35 m 28 s |
> | PubMed | - | - | - | - | 29 m 18 s | 2 d 6 h 57 m 7 s | 2 m 9 s | - | - | - |
> | Photo | 7 h 53 m 49 s | 13 h 18 m 7 s | 7 h 46 m 41 s | 3 h 0 m 23 s | 2 m 4 s | 1 h 13 m 57 s | 8 h 25 m 21 s | 17 h 32 m 58 s | 7 h 41 m 21 s | 7 h 42 m 2 s |
> | Computers | 1 d 20 h 18 m 23 s | - | - | 16 h 36 m 34 s | 19 m 8 s | 11 h 49 m 18 s | 1 m 8 s | - | - | - |
> | CoPhysics | - | - | - | - | 3 h 8 m 27 s | - | 34 m 25 s | - | - | - |
>
> ---
>
> **GCN Computation times**
>
> | Dataset | DM `Train` | DM `All` | α-BANZ `Train` | α-BANZ `All` | LOO `Train` | LOO `All` | SHAP `Train` | SHAP `All` | PCWP `Train` | PCWP `All` |
> | --- | --- | --- | --- | --- | --- | --- | --- | --- | --- | --- |
> | Citeseer | 52 m 40 s | 16 m 33 s | 51 m 6 s | 9 m 46 s | 39 s | 2 m 22 s | 53 m 35 s | 28 m 2 s | 10 m 48 s | 10 m 48 s |
> | CoraML | 3 h 53 m 11 s | 56 m 33 s | 3 h 54 m 16 s | 24 m 13 s | 1 m 7 s | 14 m 54 s | 3 h 48 m 52 s | 39 m 35 s | 17 m 39 s | 18 m 1 s |
> | PubMed | 3 h 5 m 17 s | - | 3 h 2 m 40 s | - | 50 s | 1 h 28 m 26 s | 2 m 4 s | 2 h 20 m 5 s | - | - |
> | Photo | 10 h 9 m 58 s | 14 h 24 m 27 s | 10 h 17 m 5 s | 34 m 50 s | 9 m 39 s | 5 h 22 m 32 s | 11 h 3 m 58 s | 1 h 44 m 26 s | 21 m 24 s | 21 m 25 s |
> | Computers | - | - | - | - | 19 m 12 s | 18 h 5 m 51 s | 1 m 8 s | 5 h 22 m 29 s | 27 m 38 s | 27 m 34 s |
> | CoPhysics | - | - | - | - | 17 m 23 s | - | 40 m 10 s | - | - | - |
>
> ---
>
> **GAT Computation times**
>
> | Dataset | DM `Train` | DM `All` | α-BANZ `Train` | α-BANZ `All` | LOO `Train` | LOO `All` | SHAP `Train` | SHAP `All` | PCWP `Train` |  |
> | --- | --- | --- | --- | --- | --- | --- | --- | --- | --- | --- |
> | Citeseer | 3 h 40 m 48 s | 25 m 53 s | 3 h 32 m 20 s | 19 m 36 s | 2 m 1 s | 10 m 22 s | 3 h 28 m 29 s | 1 h 0 m 43 s | 27 m 57 s | 27 m 57 s |
> | CoraML | 15 h 15 m 20 s | 1 h 17 m 7 s | 15 h 2 m 52 s | 45 m 14 s | 4 m 56 s | 1 h 11 m 2 s | 16 h 50 m 56 s | 1 h 59 m 5 s | 44 m 45 s | 44 m 45 s |
> | PubMed | 17 h 59 m 22 s | - | 17 h 46 m 60 s | - | 4 m 32 s | 10 h 56 m 16 s | 2 m 12 s | 9 h 58 m 24 s | - | - |
> | Photo | - | 12 h 30 m 27 s | - | 1 h 17 m 51 s | 44 m 25 s | 1 d 5 h 18 m 20 s | 4 d 7 h 42 m 0 s | 7 h 9 m 39 s | 41 m 51 s | 41 m 33 s |
> | Computers | - | - | - | - | 2 h 17 m 30 s | - | 1 m 3 s | 1 d 1 h 51 m 20 s | 1 h 14 m 26 s | 1 h 14 m 21 s |
> | CoPhysics | - | - | - | - | 1 h 13 m 2 s | - | 38 m 14 s | - | - | - |
>
> Experiments are still running and we’ll update the tables as they finish. We added the tables in the manuscript (see Appendix D).
>
> **W4)** While `Citeseer` and `CoraML` can be considered as “toy” datasets in the context of proposing new GNN architectures, they are still the main datasets when studying specific aspects of GNNs (adversarial robustness, explainability, oversmoothing, uncertainty quantification, etc.). Moreover, we have results for larger dataset too, like `PubMed`, `Photo`, `Computers` and `CoPhysics`. Computing existing data values on even larger graphs is computationally prohibitive. However, we agree with the reviewer that data valuation approaches need to scale and, as we pointed out in the conclusion, more effort is needed in this direction. Worth to mention, the largest graph in the only graph-specific baseline (PCW) is `CoPhysics`.

---

> > ### Comment · Reviewer_5Tw8 · 2024-11-24
> > **Thank you for the response**
> >
> > I appreciate the author's response, particularly the detailed computation cost.
> >
> > However, due to the limited novelty and scope, I'm inclined to keep my rating.

---

### Official Review · Reviewer_UsoW · 2024-11-06

**Soundness:** 2
**Presentation:** 3
**Contribution:** 2
**Rating:** 5
**Confidence:** 3

**Summary:**

This work conducts a comprehensive study in a semi-supervised transductive setting. In this setting, the paper highlights the insufficiency of the setting adopted by existing works, which ignores the influence of unlabeled nodes. The paper empirically shows that taking both labeled and unlabeled nodes into consideration will lead to a more effective valuation.

**Strengths:**

The paper summarizes and connects different valuation methods in a clear way.

The idea of considering unlabeled data in the semi-supervised transductive setting is interesting and makes much sense.

The experiments show that the overlooked data valuation methods, such as data models, perform quite well across different downstream tasks of data valuation. This is important for future research in this field.

**Weaknesses:**

My primary concern is that the motivation is not well supported by evidence. The paper starts with the motivation that we should consider the influence of unlabeled data (e.g., line 016).  Also, in line 084, the authors claim that ”we can find unlabeled nodes which have greater influence than many training nodes.” However, I could not find much empirical support in the experiments. This claim should be backed up with comparative experiments.

I find the setting part, like the “Learning signal vs. Overall signal” section, difficult to digest. It would be great if it could be expanded with concrete examples (a tiny set of nodes, some are used to train the model, some are used for inference, etc).

In Figure 1, the coloring of nodes is not used in the subsequent and might not be so meaningful in this diagram. Defining them here seems redundant and distracting. Using real numbers in this example might assist in illustrating the idea.

**Questions:**

Although methods like Data Shapley consider the coalition of nodes and can somehow capture the correlation between nodes, removing points based on individual score ranking does not seem to consider this. I wonder if this is the best we can do with the individual node scores and I am curious how the group influence/contribution plays in this context.

For Average class-wise values, what does a high in-class influence imply (e.g., in terms of prediction robustness)?

What is the intuition behind the improved performance of LOO when facing a larger distribution shift (Fig.6(b))?

This work looks into semi-supervised transductive settings, but I am curious if similar conclusions can be drawn in the context of supervised learning (e.g., which methods are more accurate than others).

---

> ### Author Response · Authors · 2024-11-20
>
> We would like to thank the reviewer for the detailed comments. We appreciate you find our work interesting and important for future investigation in the field. In the following, our response to the comments and questions.
>
> **Unlabeled data claims support** We kindly point the reviewer to Figure 9 and Figure 13 for supporting claims in favor of unlabeled node values. In particular, in Figure 9 size of nodes corresponds to the magnitude of the importance value for the test node under investigation, and the incoming edge color represent if it’s a positive relevant node (green) or negatively relevant node (red). In the main subplot, a test node (represented by squares) has a higher positive importance than training nodes (represented as circle). We updated Figure 9 with a legend to clarify the different node types. Additionally, in Figure 13, we show the nodes ranked by their values computed by different approaches. Here, we can see that test nodes (green bars) come first in the ranking than training nodes (red bars). However we agree with the reviewer that these details are not emphasized, thus we’ll make sure to clarify these in the revised version.
>
> **Learning vs Overall** We agree with the reviewer that the distinction between `Learning` and `Overall` signals might be difficult to digest. To account for this, we have include a new figure in the manuscript in appendix C.1 trying to make the distinction more clear. We remain open to any suggestion from the reviewer for a proper naming of the two settings. (See Figure 19).
>
> **Figure 1 fixes** The colors in Figure 1 indicate the importance of each node as a result of the subsets scan. We agree with the reviewer that using real number might avoid confusion with the coloring of other figures (like in Figure 9 colors represent classes). For this reason, we changed our figure accordingly, keeping the coloring in the real numbers to convey the message of positive/negative influential or uninfluential nodes. Moreover, we included a legend for the colors in the other figure (see Figure 9).
>
> **Q1)** All valuation methods we tested satisfy the linearity axiom. As pointed out in [1] (appendix E.3), the linearity allows to attribute the value of a group of datapoints as the sum of their values. For example, this is the core idea on which the brittle predictions experiment is based on — to find the smallest subset whose removal decreases the margins the most is equivalent to remove the top-k nodes according to their value. Anyhow, we agree with the reviewer that an interesting direction is going beyond the additive assumptions and develop approaches accounting for group influence.
>
> **Q2)** High in-class influence means that most important nodes are within the same class. If you think of this in terms of the margin — which is the metric we used as the utility — adding nodes from the same class increase the margins, while adding nodes from other classes decreases the margins. In terms of robustness, this means that high in-class positive influence (green values) correspond to a low error. To verify this, we plot a confusion matrix of the model’s predictions over the heatmap of the average in-class influence (see updated Figures 17-18). Indeed, we can see that the highly positive influence corresponds to the classes where the model predicts less false positives and thus is more certain about its predictions.
>
> **Q3)** The intuition behind the increasing performance of LOO when $\alpha$ increases is that the higher the $\alpha$ the more similar are the subsets wrt the ones LOO is trained on. In other words, $\alpha$ represents the probability of keeping a node in the subset. As we increase this probability the resulting subset will be more similar to just leave out one sample, namely LOO.
>
> **Q4)** We agree with the reviewer that is interesting to draw similar conclusions for the supervised learning setting. Indeed, even in the i.i.d. setting the evaluation across approaches is not carefully performed. For example, there is no work comparing data Banzhaf and Datamodels in the i.i.d. setting. Only [2] makes a connection between the two approaches but no comparison is performed. We plan to investigate this further, but in the meantime to fill this gap, we performed a preliminary experiment on Cifar10 where we compare Datamodels, data Banzhaf and our adaptation $\alpha$-Banzhaf on the data pruning experiment. We tested the attribution performance considering only 25000 subsets and training a single model for each subsets, with values for $\alpha$ equal to $0.1$ and $0.5$ (to match data Banzhaf). From our preliminary experiment, we can draw similar conclusions: data Banzhaf outperforms Datamodels. Additionally, data Banzhaf is more computationally efficient, e.g. in this experiment for $\alpha$ equals $0.1$ computing datamodels took 30 mins while computing data Banzhaf took just 6 mins; for $\alpha$ equals 0.5 computing Datamodels took 4 hours while data Banzhaf took 30 mins.

---

> > ### Author Response · Authors · 2024-11-20
> >
> > **References:**
> >
> > [1] Ilyas et al., Datamodels: Understanding predictions with data and data with predictions. Proceedings of the 39th International Conference on Machine Learning, volume 162 of Proceedings of Machine Learning Research, pp. 9525–9587. PMLR, 17–23 Jul 2022. URL https://proceedings.mlr.press/v162/ilyas22a.html.
> >
> > [2] Wang et al., Data banzhaf: A robust data valuation framework for machine learning. Proceedings
> > of The 26th International Conference on Artificial Intelligence and Statistics, volume 206 of Proceedings of Machine Learning Research, pp. 6388–6421. PMLR, 25–27 Apr 2023. URL https://proceedings.mlr.press/v206/wang23e.html.

---

### Meta-Review · Area_Chair_8puM · 2024-12-18

**Metareview:**

The paper provides an overview and analysis of methods for data point evaluations in the context of network data and graph-based predictors. It is laid out in a clear way, and well-organized. However, reviewers found that the novelty is somewhat limited, since many of the metrics themselves could be used almost off-the-shelf.

There is value in this analysis, but an overall opinion that the relative easiness of applying such valuation metrics in this setup asks for a higher bar on empirical evaluation.

**Additional Comments On Reviewer Discussion:**

I acknowledge, among other discussion outputs, the updated figures and explanations of points regarding e.g. unlabeled data. I also acknowledge the point that there is value in using classical datasets like Cora but ultimately we want to see how data valuation in large graphs can take place, and the off-the-shelf approach is not that practical as is. Even if the goal is not technical novelty, but "testing the state-of-the-art data valuation approaches in the context of graphs", for the latter there is more work to be done regarding e.g. possible heuristics and Monte Carlo approximations that may require at least a modicum of technical innovation.

---

### Decision · Program_Chairs · 2025-01-22

Reject